# p38-MAPK-mediated translation regulation during early blastocyst development is required for primitive endoderm differentiation in mice

Pablo Bora [1 ✉], Lenka Gahurova [1,2], Tomáš Mašek[3], Andrea Hauserova[1], David Potěšil[4], Denisa Jansova [2], Andrej Susor[2], Zbyněk Zdráhal[4], Anna Ajduk[5], Martin Pospíšek[3] & Alexander W. Bruce [1 ✉]

Successful specification of the two mouse blastocyst inner cell mass (ICM) lineages (the primitive endoderm (PrE) and epiblast) is a prerequisite for continued development and requires active fibroblast growth factor 4 (FGF4) signaling. Previously, we identified a role for p38 mitogen-activated protein kinases (p38-MAPKs) during PrE differentiation, but the underlying mechanisms have remained unresolved. Here, we report an early blastocyst window of p38-MAPK activity that is required to regulate ribosome-related gene expression, rRNA precursor processing, polysome formation and protein translation. We show that p38-MAPK inhibition-induced PrE phenotypes can be partially rescued by activating the translational regulator mTOR. However, similar PrE phenotypes associated with extracellular signal-regulated kinase (ERK) pathway inhibition targeting active FGF4 signaling are not affected by mTOR activation. These data indicate a specific role for p38-MAPKs in providing a permissive translational environment during mouse blastocyst PrE differentiation that is distinct from classically reported FGF4-based mechanisms.

---

[1] Laboratory of Early Mammalian Developmental Biology (LEMDB), Department of Molecular Biology & Genetics, Faculty of Science, University of South Bohemia, České Budějovice, Czech Republic. [2] Laboratory of Biochemistry and Molecular Biology of Germ Cells, Institute of Animal Physiology and Genetics, Liběchov, Czech Republic. [3] Laboratory of RNA Biochemistry, Department of Genetics and Microbiology, Faculty of Science, Charles University, Prague 2, Czech Republic. [4] Central European Institute of Technology, Masaryk University, Brno, Czech Republic. [5] Department of Embryology, Institute of Developmental Biology and Biomedical Sciences, Faculty of Biology, University of Warsaw, Warsaw, Poland. ✉email: borapa00@prf.jcu.cz; awbruce@prf.jcu.cz

During the preimplantation stage of mouse embryonic development, three blastocyst lineages emerge from two-cell fate decisions. The first decision involves spatial separation of an outer epithelium of extraembryonic and differentiating trophectoderm (TE) cells (precursors of the placenta) from a population of pluripotent inner cell mass (ICM) cells and is regulated by polarity-dependent differential Hippo pathway activation[1,2]. The second decision starts at approximately embryonic day (E) 3.25 and results in the specification and segregation of ICM cells between the pluripotent epiblast (EPI—contains the progenitor cells of the future fetus and resides deep in the ICM) and a second differentiating extraembryonic epithelium called the primitive endoderm (PrE—a polarized monolayer of superficial ICM contacting the fluid-filled cavity) by the late blastocyst (E4.5) stage[1]. Around the midpoint of blastocyst maturation (E4.0), the embryo no longer occupies a volume equivalent to that of the zygote and begins a process of nonlinear cavity expansion typified by pulsed expansions and contractions and eventual rupture of the encapsulating *zona pellucida* to permit uterine implantation (E4.5 onwards)[3].

Mouse blastocyst ICM lineage formation is under the regulation of the fibroblast growth factor (FGF)-extracellular signal-regulated kinase (ERK) signaling axis and initiates from an uncommitted population of early ICM cells coexpressing both the pluripotency-related transcription factor NANOG and the PrE-related transcription factor GATA6. Accordingly, FGF4 derived from specifying EPI progenitors acts upon precursor PrE cell populations, most significantly via FGF-receptor 1 (FGFR1), to drive precursor PrE cell differentiation[4,5]. The binding of FGF4 to FGFR1 (and to a lesser extent FGFR2) activates the ERK pathway to cause downregulation of NANOG protein expression and enables GATA6 to initiate transcription of other PrE lineage-specific genes (e.g., *Sox17* and *Gata4*, as reviewed previously[1,2]). Concomitantly, GATA6 protein expression is downregulated in EPI progenitors, resulting in a spatially randomized pattern of mutually exclusive NANOG and GATA6 expression within the ICM by the mid-blastocyst stage (E4.0)[4–10], termed the "salt & pepper" pattern of EPI and PrE progenitors[11]. This is then resolved by active cell movement, primarily mediated by migration of PrE-fated cells towards the cavity, and selective apoptosis to result in the defined ICM tissue layers of the late blastocyst (E4.5)[12,13]. Although homozygous *Fgf4* and combined *Fgfr1-Fgfr2* genetic knockouts can initiate coexpression of GATA6 and NANOG within nascent ICM nuclei, GATA6 expression is rapidly lost as the blastocyst matures (and all ICM cells solely express the EPI marker NANOG), underscoring the importance of the FGF-ERK signaling axis to PrE formation[4,5,7]. However, specified PrE survival is also dependent on platelet-derived growth factor (PDGF) signaling and activation of phosphoinositide 3-kinase (PI3K) and mechanistic target of rapamycin (mTOR)[14,15], possibly in association with activated FGFR2[16]. Recently, ICM lineage-specific specification and spatial refinement have also been reported to correlate with blastocyst cavity expansion[13], a process enabled by TE-mediated transport of water (via aquaporins), sodium ions (via sodium-potassium (Na$^+$/K$^+$) ATPases), and chloride ions (via apical Cl$^-$ channels) and by promotion of sodium-hydrogen ion exchange[17–21].

The members of the p38 family of mitogen-activated protein kinases (p38-MAPKs), comprising p38-α (MAPK14), p38-β (MAPK11), p38-γ (MAPK12/ERK6), and p38-δ (MAPK13/SAPK4), are widely studied stress-activated kinases implicated in inflammatory disease and development[22–24]. p38-MAPKs are known to be canonically activated by upstream mitogen-activated kinase kinases 3 and 6 (MAP2K3 and MAP2K6—reviewed here[22]). However, members of the FGFR family of ligand-receptor kinases are also reported to activate p38-MAPKs

(reviewed here[25]), and specific *Fgfr2* gain-of-function mutations (*Fgfr2*$^{+/Y394C}$ and *Fgfr2*$^{+/P253R}$) correlate with increased phosphorylation and activation of p38-MAPKs[26,27]. Indeed, Beare-Stevenson cutis gyrata syndrome, a genetic human condition caused by paralogous *FGFR2* mutations, results in increased phosphoactivation of the FGFR2 effector substrate FRS2[26], itself a known p38-MAPK (and ERK1/2) substrate (albeit in relation to FGF1-mediated signaling)[28,29]. p38-MAPKs are also reported to be phosphoactivated via FGFR3-mediated signaling cascades[30], and FGFR3 is robustly expressed in late blastocyst (E4.5) PrE but not EPI cells[31]. Hence, the full functional roles of p38-MAPK activity are not necessarily limited to the classically studied canonical pathways.

In the context of early mouse development, *Mapk14* genetic knockout causes aberrant placental phenotypes, impaired oxygen and nutrient transfer, and embryonic lethality[32]. p38-MAPK pharmacological inhibition (p38-MAPKi) of in vitro-cultured 2-cell-stage embryos causes 8- to 16-cell-stage developmental arrest[33], and inhibition post 16-cell stage is reported to result in failed TE differentiation and function[34,35]. Previously, we demonstrated that p38-MAPKi beginning at E3.5 (to avoid inhibiting TE specification) robustly impairs ICM differentiation towards PrE and results in late blastocyst (E4.5) embryos with increased numbers of uncommitted ICM cells (coexpressing NANOG and GATA6) but has minimal effects on EPI cell generation (defined by sole expression of NANOG). Additionally, we have reported that p38-MAPK mitigates amino acid (AA) depletion-induced oxidative stress, which is detrimental to the formation of both ICM cell lineages. However, the underlying molecular processes by which p38-MAPK activity affects ICM, specifically PrE, lineage formation, and the extent to which p38-MAPK is integrated within the existing framework of mechanistic knowledge remain largely unknown.

Using specific pharmacological-based interventions, we herein identified a minimum early mouse blastocyst window of p38-MAPK activity required for germane PrE differentiation. Employing (phospho-)proteomic and transcriptomic approaches to this window, we identified dysregulated protein translation as a major component of p38-MAPKi-induced PrE phenotypes. This dysregulation manifests as reductions in the expression of ribosome-related proteins, abnormal elevations in translation-related gene transcript levels, accumulation of unprocessed rRNA precursors, and impairment of polysome formation and protein translation. We also report that these p38-MAPKi-mediated PrE phenotypes can be partially reversed by activating the archetypal translational regulator mTOR but that a failed PrE differentiation phenotype associated with ERK pathway inhibition targeting active FGF signaling is not similarly reversed. Finally, we functionally assayed a list of candidate p38-MAPK effector proteins obtained via phosphoproteomic analyses of mouse blastocysts ±p38-MAPKi and identified MYBBP1A as a potential downstream p38-MAPK functional mediator of the PrE phenotypes described. Collectively, these data indicate a specific role for p38-MAPKs in providing a permissive translational environment that is compatible with PrE differentiation in the mouse blastocyst and is most likely distinct from classically reported FGF-based mechanisms of PrE derivation.

## Results

**Inhibition of p38-MAPK activity after E3.5 induces a PrE deficit and blastocyst cavity expansion defect.** Previously, we identified the p38-MAPK function between E3.5 and E3.75 as necessary for normal PrE differentiation by E4.5[36]. However, we wanted to further refine this required window and to assay the p38-MAPK-associated PrE phenotypes in more detail. We,

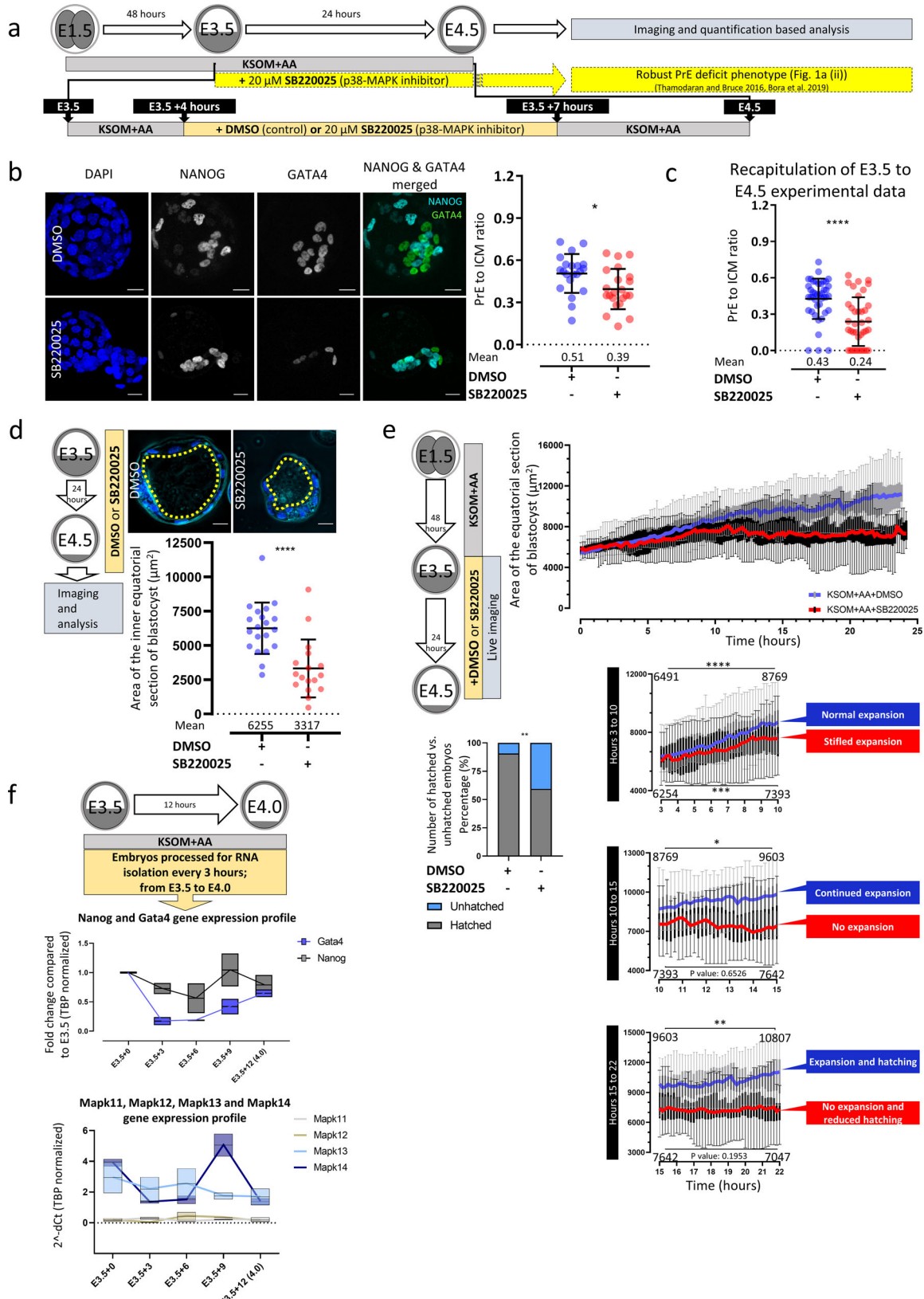

therefore, exposed blastocysts to just three hours (E3.5 + 4 to +7 hours (h)) of p38-MAPKi (20 μM SB220025[33,36,37]—dimethyl sulfoxide (DMSO) vehicle controls were also tested) and found that it was still sufficient to significantly impair PrE differentiation (Fig. 1b), albeit not as robustly as when p38-MAPKi was provided throughout the blastocyst maturation period (E3.5 to

E4.5—Fig. 1c). We previously also observed that such extended p38-MAPKi seemed to cause reduced blastocyst sizes and cavity volumes, particularly in the absence of supplemented AAs[37]. We therefore directly quantified cavity volumes in our fixed embryo samples after p38-MAPKi from E3.5-E4.5 (Fig. 1d). We observed statistically significant smaller cavities after p38-MAPKi (3316

**Fig. 1 Temporal resolution and morphological effects of inhibiting p38-MAPK signaling in developing blastocysts. a** Scheme illustrating the experimental protocol used to resolve the temporal nature of p38-MAPKi in the PrE deficit phenotype. The lineage markers NANOG and GATA4 denote EPI and PrE cells, respectively. **b** Quantification of the ratio of GATA4-positive PrE cells to total ICM cells in control ($n = 21$) and p38-MAPKi ($n = 22$) conditions from confocal images of E4.5 embryos transiently cultured with DMSO or SB220025 between E3.5 + 4 h and E3.5 + 7 h (right); the means and standard deviations are highlighted. Representative z-section blastocyst confocal microscopy projections are shown on the left (scale bars = 20 μm). **c** Quantification of the ratio of GATA4-positive PrE cells to total ICM cells in control (DMSO) ($n = 42$) and p38-MAPKi (SB220025) ($n = 38$) conditions for treatment between E3.5 and E4.5 (recapitulation of previously published observations[36,37]); the means and standard deviations are highlighted. **d** Quantification of the equatorial cavity areas (μm²) in fixed blastocysts cultured (E3.5-E4.5) in control (DMSO; $n = 20$) and p38-MAPKi (SB220025; $n = 17$) conditions. Representative confocal equatorial z-sections of each embryo group are shown, with the cavity circumference indicated (dashed yellow line) (upper panel scale bars = 20 μm). **e** Equatorial areas of control (DMSO; $n = 32$) and p38-MAPKi (SB220025; $n = 32$) blastocysts imaged in live culture from E3.5-E4.5. The sub-panels show segments of the recordings between 3 and 10 h, 10 and 15 h, and 15 and 22 h. The data are presented as box-and-whisker plots, with lines connecting the median points (blue for DMSO and red for SB220025). The numbers within the sub-panels are the mean equatorial areas. The bar graph denotes the percentages of hatching blastocysts observed at the end of the imaging period (E4.5). **f** qRT-PCR-derived expression levels (normalized to *Tbp*) of the ICM lineage markers *Nanog* and *Gata4* (upper) and the p38-MAPK paralogous genes *Mapk11-14* (lower) during early blastocyst maturation (at E3.5 + 0 h, E3.5 + 3 h, E3.5 + 6 h, E3.5 + 9 h, and E3.5 + 10 h). Data presented as box plots with the mean expression levels across the timeline expressed as line graphs.

μm²) than in controls (6255 μm²), thus quantitatively confirming our previous observations[36,37]. Extending this analysis, we employed in vitro p38-MAPKi (E3.5-E4.5) blastocyst imaging to dynamically observe cavity formation (often used to report blastocyst viability in assisted reproductive technologies[38,39]). We found that p38-MAPKi blastocysts initially expanded and displayed pulse-like cavity volume oscillations akin to those of control blastocysts in the first three hours of culture (Fig. 1e and Supplementary Videos 1–6). However, the rate of expansion in p38-MAPKi blastocysts markedly slowed in the succeeding seven hours (expansion from an average equatorial area of 6254–7393 μm²) compared to controls (expansion from 6491 to 8769 μm²—Fig. 1e (hours 3–10)). The control blastocysts then continued expanding, albeit at a reduced rate (achieving an area of 9603 μm² after 15 h), but the p38-MAPKi blastocysts did not expand (7642 μm² – (Fig. 1e (hours 10–15))). From this point onwards (ending after 24 h and at E4.5), the control embryos continued the slower rate of expansion (reaching an average size of 10,807 μm² after 22 hours of treatment), and 90.6% hatched from the *zona pellucida*; in contrast, the p38-MAPKi embryos remained unexpanded (averaging 7047 μm² in size), and in 40.6% of cases, they failed to hatch and were often associated with cavity collapse (Fig. 1e and e (hours 15–22)).

We next assayed the mRNA levels of each p38-MAPK paralog (i.e., *Mapk11*, *Mapk12*, *Mapk13*, and *Mapk14*, plus the ICM lineage markers *Nanog* and *Gata4*) during early blastocyst maturation in order to correlate the expression of these genes with the observed p38-MAPKi-sensitive period. The results revealed that *Mapk13* and *Mapk14* (encoding p38-δ and p38-α isoforms, respectively) were the only robustly expressed p38-MAPK transcripts in early blastocysts. Moreover, *Nanog* and *Gata4* mRNA levels increased when *Mapk14* transcript expression peaked between the assayed E3.5 + 6 h and E3.5 + 9 h time points (Fig. 1f). This period partly overlaps with the minimum window of p38-MAPKi sensitivity regarding PrE differentiation (Fig. 1b) and the transition from impaired to blocked cavity expansion (Fig. 1e). These data implicate the *Mapk14* gene as the prime phenotype-mediating candidate. However, they do not rule out the other paralogs, especially *Mapk13*, given that *Mapk14*-null embryos are recovered in normal Mendelian ratios until E10.5[32], suggesting some level of functional redundancy and/or p38-MAPK maternal protein inheritance. Interestingly, *Mapk14* genetic knockout alone is embryonic lethal[32,40] (unlike knockout of the three other paralogs[41,42]).

Collectively, these introductory data confirm that p38-MAPK (s) play a transient role during early blastocyst maturation, which subsequently ensures appropriate PrE differentiation and

blastocyst cavity expansion. The most important period is between the third and tenth hours after E3.5 (but can be refined to a minimum 3-h window).

## p38-MAPK inhibition in developing blastocysts affects the translation machinery.

To gain a molecular understanding of how p38-MAPK activity exerts its effect on PrE differentiation, we employed global analyses of (phospho-)proteomic changes induced by p38-MAPKi. Using mass spectrometry (MS)-based approach, we assayed the total proteomes and phosphoproteomes of cultured mouse blastocysts exposed to seven hours of p38-MAPKi (spanning the confirmed minimum developmental sensitive window—E3.5 + 2 to +9 h). We acknowledged that the output would comprise data from both TE and ICM lineages but reasoned that identification of differentially expressed (phospho-) proteins (between the p38-MAPKi and control conditions) might be able to provide important mechanistic insight and possibly (in the case of the phosphoproteome) identify direct p38-MAPK effector substrates. We detected 77 downregulated and 53 upregulated proteins associated with p38-MAPKi (Fig. 2c and Supplementary Data 1). Subsequent ontological analysis revealed that the only significantly enriched terms were directly related to "translation" or "ribosome assembly" (hereafter collectively referred to as "translation"—Fig. 2b and Supplementary Data 1), which together comprised 28 of the differentially expressed proteins (Fig. 2c). Of these, 23 were downregulated, and 5 were upregulated (Fig. 2c). Hence, p38-MAPKi during early blastocyst maturation dysregulates a specific cohort of translation-associated proteins, mostly resulting in their reduced expression.

A similar investigation of acquired phosphoproteomic data revealed 156 significantly and differentially enriched phosphopeptides (exhibiting ≥ 1.5-fold differential abundance; 99 enhanced and 57 depleted). Collectively, these phosphopeptides represented 110 individual proteins and corresponded to 59 proteins with solely enhanced phosphorylation, 41 proteins with solely depleted phosphorylation, and 10 proteins with both enhanced and depleted levels of phosphorylation at distinct sites (detailed in later sections). Of the 51 identified proteins for which at least one depleted phosphopeptide was detected (6 proteins contained two distinct depleted phosphopeptides), we identified 14 listed in the *PhosphoSitePlus* database[43] as confirmed or predicted p38-MAPK substrates (representing all four p38-MAPK isoforms in both human and mouse models—Supplementary Data 2). Twenty-four of the 57 depleted phosphopeptide sequences (42.1%) contained the consensus [S/T]P dipeptide phosphorylation motif typical of the proline-directed mitogen-activated kinase superfamily in general[44] (Supplementary Data 2),

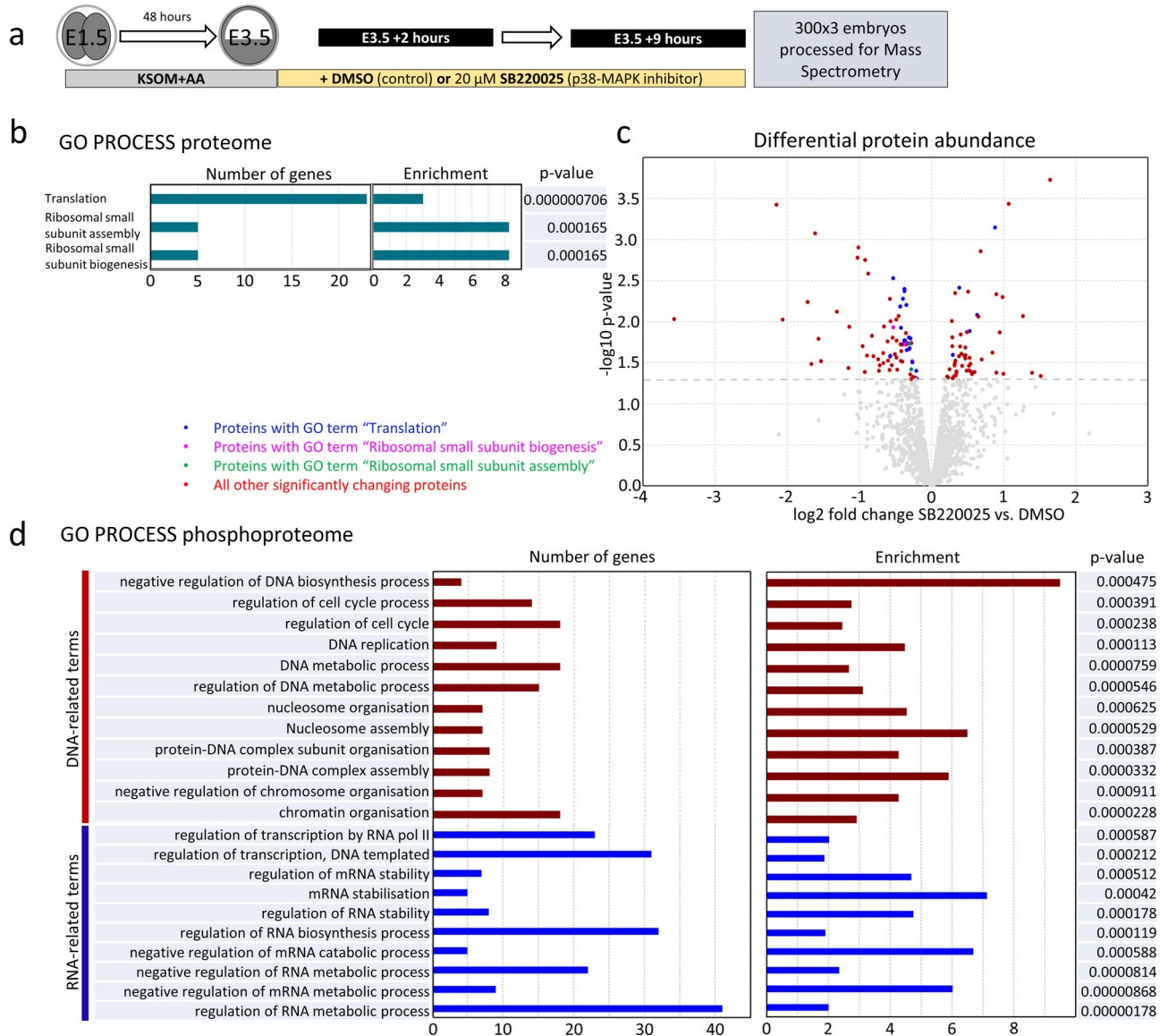

**Fig. 2 Proteomic and phosphoproteomic analyses of the effects of p38-MAPKi in developing blastocysts. a** Experimental design of sample collection for control (DMSO; $n = 3$ (300 embryos per repeat)) and p38-MAPKi (SB220025; $n = 3$ (300 embryos per repeat)) blastocysts after 7 h of chemical exposure (at E3.5 + 9 h) prior to mass spectrometric analysis of the (phospho-)proteome (300 blastocysts per condition were tested in biological triplicates). **b** The most statistically significantly enriched GO terms for the differentially detected proteins between control and p38-MAPKi blastocysts. **c** Volcano plot of the detected differentially expressed proteins associated with p38-MAPKi. Proteins associated with the indicated enriched GO terms are highlighted in blue (translation), purple (ribosomal small subunit biogenesis), and green (ribosomal small subunit assembly) (note that there is overlap—see Supplementary Data 1); the proteins associated with other terms are in red, while the proteins that were not statistically significantly changed are in gray. **d** Statistically significantly enriched GO terms for the differentially detected phosphopeptides between control and p38-MAPKi blastocysts.

suggesting that they may be direct p38-MAPK substrates. An ontological analysis of all the phosphoproteomic screen-identified genes (comprising enhanced and/or depleted phosphopeptides) revealed enrichment of RNA regulation and stability terms (Fig. 2d and Supplementary Data 1) that were consistent with translation-related terms identified for the general proteome (Fig. 2b). Collectively, these findings suggest that p38-MAPKi may impair general protein synthesis. We also noted terms related to cell cycle regulation, DNA replication, and chromatin modification (Fig. 2d and Supplementary Data 1), which was perhaps unsurprising given the known wide range of p38-MAPK effectors[23].

To complement the (phospho-)proteomic analyses and provide an extra layer of mechanistic insight, we conducted

transcriptomic analyses of p38-MAPKi-treated blastocysts at three different early blastocyst time points (E3.5 + 4 h, +7 h, and +10 h of p38-MAPKi—Fig. 3a). We detected 34 differentially expressed transcripts at the +4 h time-point, 1240 at the +7 h time-point, and 480 at the +10 h time-point in the treated blastocysts compared to the control blastocysts; only 10 differentially expressed mRNAs overlapped all three inhibition regimes (Fig. 3b and Supplementary Data 1). These data indicated the occurrence of large changes in the p38-MAPK-regulated transcriptome that are maximally centered around the E3.5 + 7 h time-point (Fig. 3b and Supplementary Data 1), specifically a cohort of differentially expressed transcripts comprising both upregulated and downregulated genes (Supplementary Data 1). Hierarchical clustering of this gene cohort across all assayed

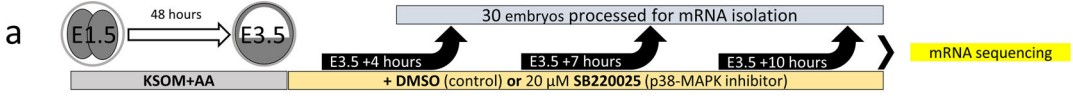

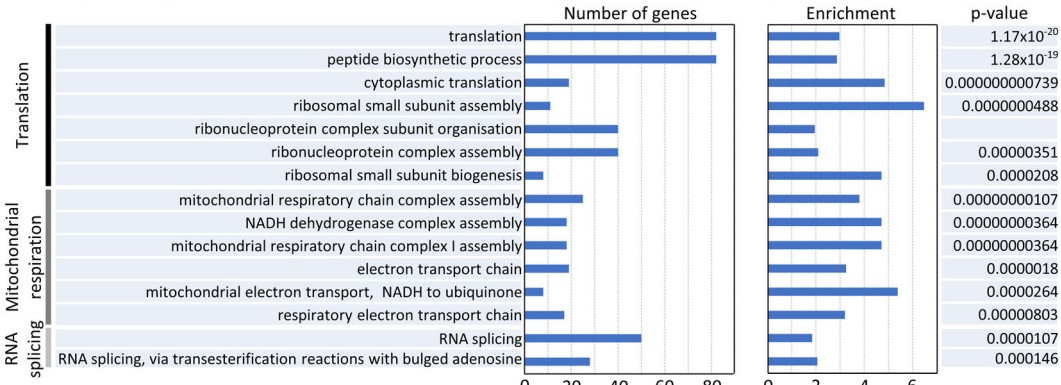

**b** Differentially expressed genes; DMSO vs SB220025 (DESeq2)

**c** Genes significantly differentially expressed in E3.5 +7 hours (numbered are clusters with 20+ genes)

**d** GO analysis of top 15 terms from E3.5 +7 hours (excluding generalised terms e.g. cellular processes)

| | Number of genes | Enrichment | p-value |
|---|---|---|---|
| translation | | | $1.17 \times 10^{-20}$ |
| peptide biosynthetic process | | | $1.28 \times 10^{-19}$ |
| cytoplasmic translation | | | 0.000000000739 |
| ribosomal small subunit assembly | | | 0.0000000488 |
| ribonucleoprotein complex subunit organisation | | | |
| ribonucleoprotein complex assembly | | | 0.00000351 |
| ribosomal small subunit biogenesis | | | 0.00000208 |
| mitochondrial respiratory chain complex assembly | | | 0.00000000107 |
| NADH dehydrogenase complex assembly | | | 0.00000000364 |
| mitochondrial respiratory chain complex I assembly | | | 0.00000000364 |
| electron transport chain | | | 0.0000018 |
| mitochondrial electron transport, NADH to ubiquinone | | | 0.0000264 |
| respiratory electron transport chain | | | 0.00000803 |
| RNA splicing | | | 0.0000107 |
| RNA splicing, via transesterification reactions with bulged adenosine | | | 0.000146 |

(Translation / Mitochondrial respiration / RNA splicing)

**e** RNAseq data of genes demonstrating proteome level downregulation

treatments and time points identified three clusters of coregulated genes (Fig. 3c—a similar clustering of differentially regulated genes at E3.5 + 4 h and E3.5 + 10 h is described in Supplementary Fig. 1a, b and Supplementary Data 1). Cluster I represents transcripts that were downregulated after p38-MAPKi at the E3.5 + 4 h time point, were maximally repressed at E3.5 + 7 h but

returned to levels equivalent to the control levels by E3.5 + 10 h. The transcripts in cluster III demonstrated a similar but inverted trend. Namely, at E3.5 + 4 h, p38-MAPKi initiated an increase in transcript expression that peaked by E3.5 + 7 h, but the expression returned to control levels by E3.5 + 10 h. Cluster II comprised similarly enhanced transcripts but was distinguished

**Fig. 3 Temporal transcriptomic analysis of the effects of p38-MAPKi during early blastocyst maturation. a** Experimental design for transcriptome sample collection for the control (DMSO) and p38-MAPKi (SB220025) conditions from E3.5 to +4, +7, or +10 h (biological duplicates of 30 blastocysts per condition were tested per time point). **b** Venn diagram showing the numbers of differentially expressed mRNAs (as determined by DESeq2 analysis) and their overlap between control and p38-MAPKi blastocysts at the three selected time points (i.e., E3.5 + 4 h, +7 h, and +10 h). **c** Hierarchical clustering heatmap depicting the expression of mRNAs that were significantly changed by p38-MAPKi at the +7 h time point and the transcript levels of those same genes at the +4 and +10 h time points. The mRNAs formed three distinct expression clusters. **d** Top 15 terms identified in GO enrichment analysis for the p38-MAPK-regulated transcriptome at the +7 h time-point (excluding generalized terms, e.g., cell). **e** Hierarchical clustering heatmap of genes originally identified as downregulated at the protein level and associated with translation-related GO terms after p38-MAPKi (Fig. 2b, c) at the three assayed early blastocyst ±p38-MAPKi time points.

from cluster III in that its transcripts did not return to control levels by E3.5 + 10 h (for individual genes from all three clusters in Fig. 3c see Supplementary Data 1). Thus, p38-MAPKi caused large-scale transcriptomic changes that peaked at the E3.5 + 7 h time point and (with the exception of the enhanced gene transcripts of cluster II) largely returned to levels observed in control blastocysts by the E3.5 + 10 h time point via compensation. Notwithstanding such compensation, it is important to recall that p38-MAPKi throughout this same period (or in the identified minimum window) is sufficient to ultimately impair PrE differentiation (Fig. 1 and previous publications[36,37]), potentially due to impaired mRNA translation (see below). Interestingly, we did not observe statistically significant mRNA expression changes of the same differentially regulated genes among the three control conditions (+DMSO at E3.5 + 4 h, +7 h, and +10 h), indicating these genes are not normally subject to dynamic regulation during this period. Remarkably, an ontological analysis of enriched terms associated with such p38-MAPKi-sensitive transcripts again identified translation-related terms as the most significantly enriched (Fig. 3d and Supplementary Data 1), which was mostly reflected by upregulated expression (cluster III, Fig. 3c). Of these 82 differentially expressed translation-related mRNAs, 79 were upregulated after p38-MAPKi (Supplementary Data 1—see tab labeled Fig. 3d, e). Interestingly, 13 of those 79 transcripts (mainly representing small ribosomal subunit proteins) were also identified as downregulated in the general proteome (Fig. 3e and Supplementary Data 1). These data suggest that the accumulation of translation-related mRNAs caused by p38-MAPKi (peaking at E3.5 + 7 h) may reflect impaired ribosomal assembly and/or functional protein translation. Moreover, they also indicate a potential positive feedback mechanism culminating in the observed transient increases in translation/ribosome-related gene transcript levels (Fig. 3c, d). Other ontological terms enriched at E3.5 + 7 h were connected to mitochondrial respiration (also upregulated—cluster III, Fig. 3c) and splicing (associated with both upregulated and downregulated transcripts—clusters I and III, respectively, Fig. 3c).

Overall, the combined results of the (phospho-)proteomic and transcriptomic analyses point to the regulation of translation as a primary function of p38-MAPK activity during early blastocyst maturation, which may help create a permissive state for germane PrE differentiation.

**p38-MAPK inhibition reduces active translation during early blastocyst maturation.** Informed by the (phospho-)proteomic and transcriptomic screens, we hypothesized that p38-MAPKi-induced PrE phenotypes are associated with disrupted protein translation during early blastocyst maturation. To test this hypothesis, we assayed the effect of p38-MAPKi on a translation during a 10-hour period (starting from E3.5) utilizing an O-propargyl-puromycin (OPP) incorporation assay that permits direct fluorescent labeling and quantification of nascent translated polypeptides (Fig. 4a). As expected, we observed nearly 50% less global protein synthesis under p38-MAPKi conditions than under control conditions (Fig. 4b). We also verified the observed reduction in active translation (in the same time period) using a polysome profiling approach, adopting our recently reported protocol of scarce-sample polysome (SSP) profiling[45]. This involved obtaining ten fractions representing individual ribosomal subunits, monosomes, and polysomes from 10 blastocysts per control or p38-MAPKi condition in quadruplicate. The fractions were then processed for total RNA extraction and cDNA synthesis and analyzed by qRT-PCR to specifically assay 18S and 28S rRNA levels. We observed shifts in the distributions of both 18S and 28S rRNA abundance towards SSP fractions indicative of free rRNAs or associations with individual ribosomal subunits/monosomes (fractions 1–5) versus denser fractions indicative of polysomes (fractions 6–10) after p38-MAPKi (Fig. 4c). We also expressed the results as calculated polysome-to-non-polysome ratios (with increased ratios indicative of enhanced active translation—Fig. 4d). Interestingly, the reduction in the polysome-to-non-polysome ratio was greater for 28S rRNA than for 18S rRNA (Fig. 4d). On the basis of these combined data, we conclude that active translation is reduced in intact early mouse blastocysts under p38-MAPKi conditions.

**rRNA processing in early blastocysts is impaired by p38-MAPK inhibition.** Due to the general dysregulation of translation-related gene expression in early blastocysts and the confirmed impairment of active protein translation caused by p38-MAPKi, we sought to identify other contributory mechanisms underlying the associated PrE differentiation phenotypes. Upon searching the list of potential p38-MAPK effectors from the phosphoproteomic screen, we identified MYBPP1A (from which 6 individual differentially expressed phosphopeptides had been detected under p38-MAPKi conditions; expanded below). MYBPP1A interacts with HTATSF1, a factor known to be involved in transcriptional and post-transcriptional regulation/processing of rRNA and mRNA and regulation of protein synthesis and a confirmed modulator of in vivo embryonic pluripotency (during the transition from pre- to postimplantation development)[46]. Thus, we assayed the effect of p38-MAPKi on 45S pre-rRNA processing during the early blastocyst developmental window (E3.5 + 10 h—employing the same qRT-PCR primers used in the HTATSF1 report of Corsini et al.[46]). This revealed that p38-MAPK regulates rRNA processing (Fig. 4e, f). Specifically, we observed enhanced levels of PCR amplicons indicative of impaired processing. For example, the levels of the internal transcribed spacer 2 (ITS2) region, normally removed from processed 45S pre-rRNA, were over 2-fold increased by p38-MAPKi. Similarly, the levels of 5′ external transcribed spacer (ETS) amplicons were increased, as were those of regions representative of the overlap between ITS2 and the mature 28S rRNA coding sequence, plus the overlapping region between the ETS and 18S rRNA; all of these changes indicated impaired 45S pre-rRNA processing. Upon assaying regions of the 45S pre-rRNA corresponding to mature (appropriately processed) rRNA

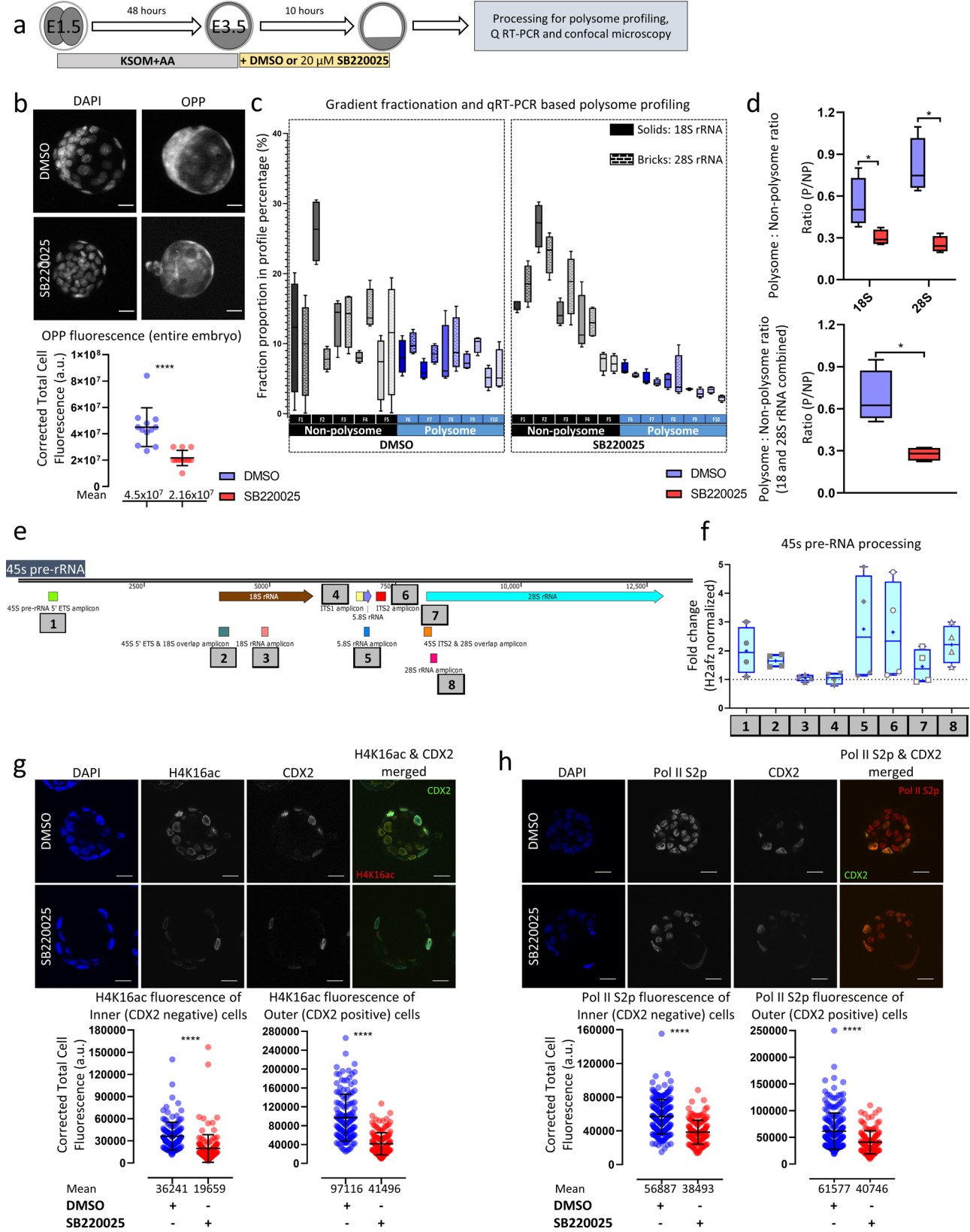

transcripts, we also observed increases in 28S and 5.8S rRNA levels, presumably reflecting unprocessed 45S pre-rRNA intermediates. However, similar assays of the 18S rRNA transcript were not associated with increased abundance, suggesting that steady-state levels of processed 18S rRNA were not significantly affected. This was also reflected in the unchanged levels of ITS1

and distinguished p38-MAPKi-derived data from studies on *Htatsf1* genetic knockouts, in which ITS levels were increased (and ETS levels were reduced)[46]. Notwithstanding such differences, our data confirm a role for p38-MAPK in regulating 45S pre-RNA processing in early blastocysts that can feed forward to influence active protein translation.

**Fig. 4 Effects of p38-MAPKi on active translation, rRNA processing, and the transcriptional landscape in developing blastocysts. a** Experimental design for analysis of the roles of p38-MAPK activity in developmental translation and transcription in maturing blastocysts (between E3.5 and E3.5 + 10 h). **b** Representative confocal z-projection micrographs (upper) of E3.5 + 10 h blastocysts in control (DMSO; n = 12) and p38-MAPKi (SB220025; n = 12) conditions depicting cell nuclear staining (DAPI) and de novo translation (after OPP incorporation); scale bar = 20 μm. The scatterplot shows the corrected total cell fluorescence (CTCFs) of OPP incorporation between the two conditions (lower); the means and standard deviations are highlighted. **c** qRT-PCR analyses quantifying 18S and 28S rRNA levels in each of ten SSP profiling fractions representative of polysome-associated (F6-F10) and non-polysome-associated (F1-F5) rRNA transcripts obtained from both control and p38-MAPKi blastocysts (E3.5 + 10 h). Box-and-whisker plots represent the proportion of the overall amount of rRNA transcript detected for a given fraction. The data were generated from four sets of 10 embryos each, per condition. **d** Ratios of polysome- to non-polysome-quantified rRNAs (as derived from the data in **c**) for 18S and 28S separately (top panel) and combined (bottom panel). The data are presented in interleaved box-and-whisker plots, with the whiskers depicting the minimum and maximum values and the boxes including the medians and the 25th and 75th percentiles. **e** Schematic representation of 45S pre-rRNA highlighting the qRT-PCR amplicons assayed and quantified to analyze rRNA processing status ±p38-MAPKi (E3.5 + 10 h). **f** Box-and-whisker plots (25th to 75th percentile in box and minimum and maximum values indicated by whiskers, with median and mean values marked by a horizontal bar and + sign, respectively; n = 4) representing qRT-PCR-quantified fold-changes of the different specific amplicons (normalized against *H2afz* cDNA) in p38-MAPKi versus control blastocysts (DMSO). **g** Representative single z-stack confocal micrographs (upper) of E3.5 + 10 h blastocysts under control (DMSO; n = 13 embryos) and p38-MAPKi (SB220025; n = 15 embryos) conditions. Individual channel micrographs for DAPI (blue pan-nuclear stain; shows the total number of cells), H4K16ac (grayscale; a posttranslational histone chromatin mark associated with transcriptional activity), and CDX2 (grayscale; marks outer TE cells) plus a merged H4K16ac (red) and CDX2 (green) image are shown (scale bar = 20 μm). Scatterplots (lower) quantifying the H4K16ac (CTCF) level per nucleus in control (DMSO) and p38-MAPKi (SB220025) blastocysts differentiated between inner (n = 129 for DMSO and n = 143 for SB220025) and outer (n = 136 for DMSO and n = 145 for SB220025) cell populations are also shown; the means and standard deviations are depicted. **h** Similar to **g** but describes the expression levels of the RNA pol II S2p protein. Quantification was performed for inner cells (n = 187 for DMSO and n = 135 for SB220025) and outer cells (n = 209 for DMSO and n = 123 for SB220025) from n = 11 embryos in DMSO and n = 14 embryos in SB220025 conditions, with the means and standard deviations shown.

**Early blastocyst transcription is attenuated by p38-MAPK inhibition**. In the context of peri-implantation embryonic development and in vitro pluripotency, recent reports have drawn important correlations between general translational status and permissive transcriptional landscapes. We investigated whether the observed p38-MAPKi-mediated defects in protein synthesis were also associated with reduced transcription (Fig. 4g, h). Consistently, we observed reductions in the levels of histone H4-lysine 16 acetylation (H4K16ac, a posttranslational mark of actively transcribed chromatin[47]) in both inner and outer cells, as demarcated by the expression of CDX2 (an outer cell TE marker[48]), in p38-MAPKi blastocysts (E3.5 + 10 h). Similarly, immunofluorescence staining for actively elongating RNA polymerase II (using an antibody recognizing RNA polymerase II phosphorylated on serine-2 of the C-terminal domain—RNA pol II S2p[49]) showed marked decreases.

Hence, p38-MAPKi during early blastocyst maturation, known to negatively affect PrE differentiation (as described above[36,37]), is collectively associated with impaired translation, polysome formation, rRNA processing, and a chromatin landscape suggesting reduced active transcription.

**p38-MAPK and mTOR have partially overlapping roles in regulating blastocyst development**. Although p38-MAPK activity is reported to regulate protein synthesis in some contexts[50], mTOR-mediated translation regulation is the most extensively studied pathway, particularly with regard to development[51]. Cancer biology studies have indicated a possible convergence of both pathways that positively regulates eIF4E-sensitive cap-dependent mRNA translation[52,53]. Given the prominent role mTOR plays in translation regulation and a recent report describing blastocyst diapause caused by attenuated translation and transcription after mTOR inhibition[54], we investigated our observed p38-MAPKi phenotypes in conjunction with mTOR activity. First, we compared p38-MAPKi and mTOR inhibition phenotypes. Early blastocysts (E3.5) were cultured in the presence of DMSO (control), TORIN1 (an mTOR inhibitor[55]), and SB220025 (a p38-MAPK inhibitor) for 24 h and immunofluorescently stained for ICM cell lineage markers (NANOG for EPI and GATA4 & GATA6 for PrE—

Fig. 5a). mTOR inhibition, like p38-MAPKi, caused a statistically significant increase in the proportion of NANOG- and GATA6-coexpressing cells within the ICM (Fig. 5b–d, p); NANOG and GATA6 coexpression is a hallmark of unspecified cells[11] and a potentially diapaused state. Likewise, the numbers of PrE cells (defined by sole GATA4 expression—Fig. 5b–d, n, o) in both p38-MAPKi and mTOR-inhibited blastocysts were also significantly reduced, confirming our previous p38-MAPKi observations[36,37] and the report of diapause in mTOR-inhibited blastocysts[54]. However, despite the similarities, mTOR-inhibited blastocysts had fewer EPI (and overall ICM) cells than p38-MAPKi and control embryos (defined by sole NANOG expression—Fig. 5b–d, l, m), indicating that mTOR-inhibited blastocysts are diapaused at the early blastocyst stage and characterized by an unspecified ICM. Conversely, p38-MAPKi blastocysts did have specified EPI cells and exhibited impairment only in PrE formation (retaining a population of unspecified ICM cells), indicating that the p38-MAPKi phenotype and the associated translational defects are squarely centered on PrE differentiation. Therefore, p38-MAPKi does not merely phenocopy mTOR inhibition-induced diapause (as further supported by the inability of p38-MAPKi, unlike mTOR inhibition[54], to sustain long-term in vitro embryo survival after E4.5—Supplementary Fig. 2). However, the data also do not exclude possible p38-MAPK and mTOR functional convergence in the limited context of PrE differentiation.

Therefore, we next tested whether the p38-MAPKi-mediated PrE phenotypes could be ameliorated by the pharmacological activation of mTOR. We utilized the potent mTOR activator MHY1485[56], which, when provided in isolation to maturing blastocysts (E3.5-E4.5), did not affect the development or derived ICM cell fate (Fig. 5a, e, j–q). However, the number of GATA4-expressing PrE cells was statistically greater when MHY1485 was administered in combination with p38-MAPKi than when p38-MAPKi was administered alone (but was not at the levels seen in controls—Fig. 5a, b, f, n). There was also a reduced contribution of the otherwise typical p38-MAPKi-induced population of unspecified ICM cells (Fig. 5p), but there was no change in EPI cell numbers (also reflected by the calculated PrE-to-ICM ratios—Fig. 5m, o). Intriguingly, mTOR activation under p38-MAPKi

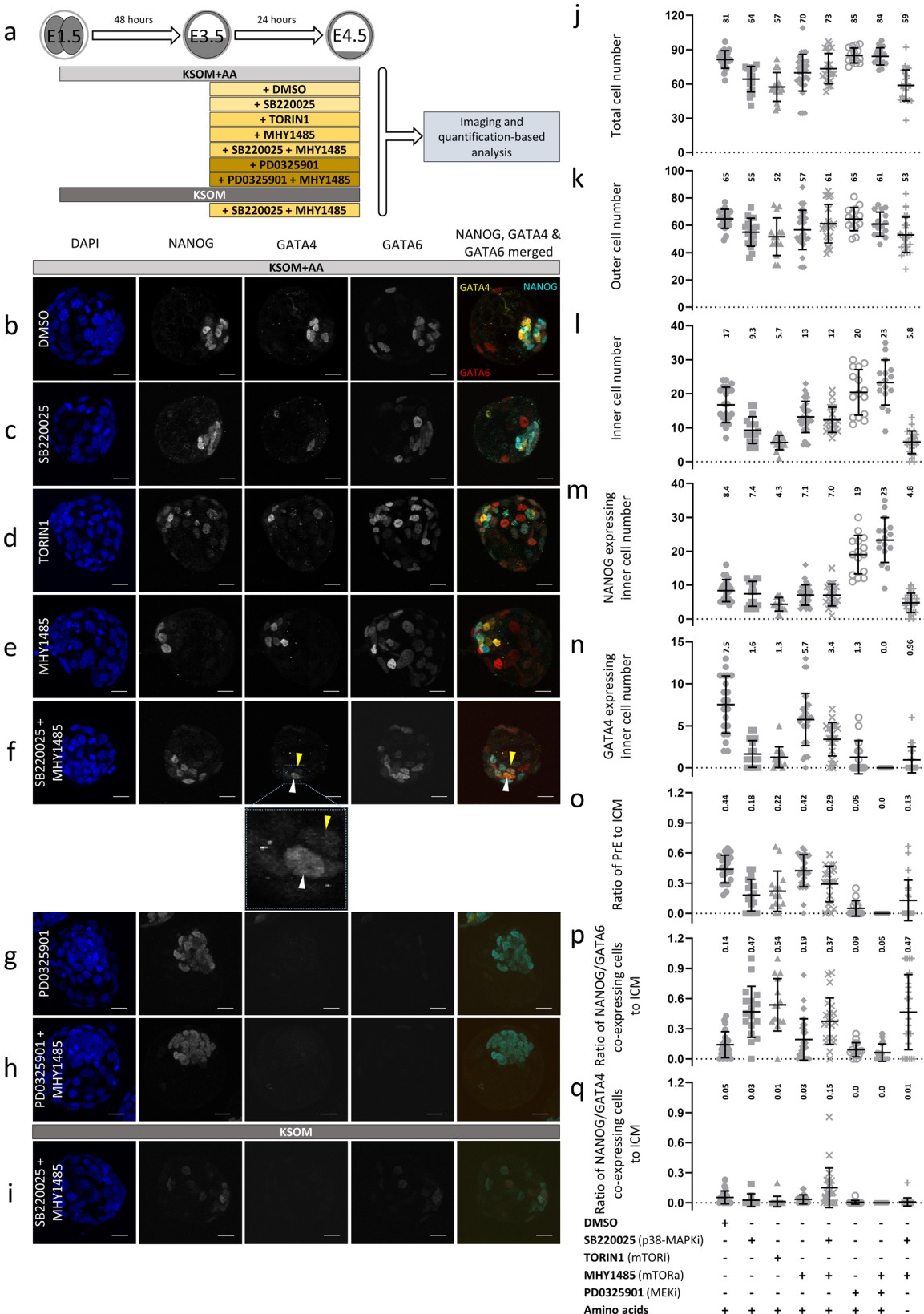

also resulted in coexpression of NANOG and GATA4 by a statistically significant population of ICM cells, which was seldom observed in all other conditions, including control conditions (Fig. 5f, q). These data show that p38-MAPKi-mediated deficits in PrE differentiation associated with a unique population of unspecified PrE progenitors can be partially overcome by

activation of mTOR without an accompanying decrease in NANOG expression.

The FGF4-ERK signaling cascade is considered the archetypal ICM cell fate-determining mechanism of mouse blastocysts[57]. Therefore, we also assayed whether PrE differentiation phenotypes induced by inhibition of the ERK pathway (via targeting of

**Fig. 5 Functional interplay between mTOR and p38-MAPK during blastocyst maturation and PrE differentiation. a** Experimental schematic for analysis of the potential functional overlap of mTOR and p38-MAPK with regard to PrE differentiation (via assay of ICM lineage marker protein expression) during blastocyst maturation. E3.5-stage blastocysts were cultured for 24 h in medium (±AAs) containing the indicated pharmacological supplements (vehicle control: DMSO; p38-MAPK inhibitor: SB220025; mTOR inhibitor: TORIN1; mTOR activator: MHY1485; MEK1/2 inhibitor: PD0325901) or combinations thereof. **b–i** Representative confocal z-series projections of blastocysts cultured under the conditions described in **a** and stained with DAPI (blue) and for NANOG, GATA4, and GATA6 protein expression (cyan, yellow and red in the provided merged image); scale bar = 20 μm. **b** DMSO (n = 21), **c** p38-MAPKi, SB220025 (n = 17), **d** mTOR inhibition, TORIN1 (n = 15), **e** mTOR activation, MHY1485 (n = 32), **f** p38-MAPKi and mTOR activation, SB220025 + MHY1485 (n = 22), **g** MEK inhibition, PD0325901 (n = 15), **h** MEK inhibition and mTOR activation, PD0325901 + MHY1485 (n = 16) and **i** p38-MAPKi and mTOR activation, SB220025 + MHY1485 in KSOM not supplemented with AAs (n = 24). In **f** and the expanded view, the white arrowheads highlight differentiated PrE cells expressing GATA4 (and GATA6) but not NANOG, whereas the yellow arrowheads show ICM cells coexpressing GATA4 and NANOG (which were comparatively enriched in this condition). **j–q** Scatterplots showing the total, outer and inner blastocyst cell numbers (as indicated by DAPI staining) and the numbers of NANOG-, GATA4- and GATA6-expressing ICM cells under the culture conditions described in **a** (with the means and standard deviations highlighted). **j** Total cells; **k** outer cells; **l** inner cells; **m** NANOG-expressing inner cells; **n** GATA4-expressing inner cells; **o** the ratio of GATA4-positive, NANOG-negative PrE cells to total ICM cells; **p** the ratio of NANOG- and GATA6-coexpressing cells to total ICM cells; and **q** the ratio of NANOG- and GATA4-coexpressing cells to total ICM cells.

MEK1/2, also known as MAP2K1/2, with PD0325901[58]) were sensitive to mTOR activation. While ERK pathway inhibition (E3.5-E4.5) resulted in the well-defined phenotype of largely pan-ICM expression of NANOG alone, PrE was not rescued by additional activation of mTOR; rather, PrE inhibition was augmented (Fig. 5g, h, j–q). We conclude that the role of p38-MAPK in ICM cell fate specification via translation regulation is functionally and mechanistically distinct from that of the described FGFR-ERK pathway.

We have previously reported that p38-MAPK also acts to counter AA depletion-induced oxidative stress in mouse blastocysts, as revealed by an exacerbation of ICM cell fate phenotypes affecting both EPI and PrE when p38-MAPKi is combined with an absence of exogenous AA in the medium[37]. We hypothesized that activating mTOR may also alleviate such enhanced effects (E3.5-E4.5). However, unlike those cultured in the presence of supplemented AA, blastocysts cultured under such conditions did not demonstrate any induced inner cell GATA4 expression or increased inner cell numbers, nor were there any increases in the numbers of EPI cells expressing NANOG alone (Fig. 5i, j–q). We conclude that the previously identified role of p38-MAPK in counteracting oxidative stress is distinct from that described here in relation to mTOR activity and translation.

**MYBBP1a has a prominent role in preimplantation embryonic development.** Having uncovered a role for p38-MAPK in regulating translation and PrE differentiation in early blastocysts, we sought to identify and functionally verify contributing p38-MAPK effectors. We anticipated that our list of p38-MAPKi-induced differentially expressed phosphoproteins would include strong candidates (Fig. 2d and Supplementary Data 1). We filtered out detected phosphopeptides (and associated phosphoproteins) demonstrating <1.5-fold changes in phosphorylation and retained those exhibiting <1.3-fold changes in general protein abundance (Fig. 6a and Supplementary Data 1). We further refined the list according to the published literature (identifying proteins/genes shown or hypothesized to regulate pluripotency, differentiation, and protein translation) and selected six genes (highlighted in Fig. 6a—associated with one or more differentially phosphorylated peptides/proteins) for functional assays in embryonic clonal loss-of-function experiments (Dnmt1, Kdm2B, Ybx1, Mybbp1a, Rpl22l1, and Nolc1). All genes were robustly and equally expressed at the mRNA level in published screens of mouse blastocyst ICM and TE lineages, implying that their functional regulation is likely to be post-transcriptional[59,60] (Supplementary Fig. 3). We employed candidate-specific siRNA constructs (or non-targeting controls) to knockdown gene

expression in marked preimplantation embryo cell clones. The appropriate siRNA and a recombinant mRNA encoding a histone H2B-RFP fusion reporter were microinjected into one cell of 2-cell-stage (E1.5) embryos to create a fluorescently marked and traceable cell clone composing half the embryo. The embryos were cultured to the late blastocyst (E4.5) stage and immunofluorescently stained for ICM cell fate markers (NANOG for EPI and GATA4 for PrE). Two-cell-stage embryos microinjected in both blastomeres and cultured to the late blastocyst stage were subjected to qRT-PCR to assess specific candidate gene transcript knockdown (Fig. 6b). We found that targeting Mybbp1a had the most substantial developmental impact among the six shortlisted genes, despite relatively modest (61%) transcript knockdown efficiency (Fig. 6c); the effects of knockdown of the other candidate genes are summarized in the Supplementary Information (Supplementary Data 2). MYBBP1A is a predominantly nucleolar protein reported to negatively regulate RNA polymerase I-dependent rRNA gene transcription, co-transcriptional processing, and ribosome biogenesis[61,62]; p53 tetramerization, cell senescence, and apoptosis[63,64]; and Hoxb2 gene transcription[65]. Additionally, homozygous Mybbp1a-null blastocysts resulting from crosses of heterozygous Mybbp1a-null mice cannot develop[66], implying an important but as-yet-undefined role(s) of Mybbp1a in preimplantation development. Clonal analysis of marked Mybbp1a-knockdown cells revealed a 19% contribution to the late blastocyst (E4.5)-stage ICM and a 40% contribution to the outer TE, compared to 50% and 51% contributions, respectively, in the control group (Fig. 6d–f and Supplementary Fig. 4). However, such reduced contributions were also associated with an overall reduction in total clone size compared to that of equivalent microinjected clones of control siRNA-treated embryos (on average comprising 16.57 ± 6.20 versus 39.33 ± 4.71 cells, respectively) or noninjected sister clones (Fig. 6f). Despite this reduction, Mybbp1a-knockdown cells that populated the ICM rarely, if at all, contributed to GATA4-expressing PrE (representing an average of only 0.11 ± 0.42 cells per embryo or 4% of the total PrE population). Conversely, such clones readily contributed to NANOG-positive EPI (an average of 1.29 ± 1.56 cells per embryo or 29% of the total EPI cell number). In contrast, in control siRNA-treated embryos, the contributions of both microinjected and non-microinjected clones between the ICM and outer TE and between the EPI and PrE lineages were always observed to be statistically equal (Fig. 6d–f and Supplementary Fig. 4). Interestingly, clonal Mybbp1a-knockdown mildly affected the non-microinjected clone, causing small and statistically nonsignificant reductions in total, outer and inner cell numbers. There were, however, reduced numbers of GATA4-positive cells (on average, 2.82 ± 3.83 cells versus 5.2 ± 2.77 cells in the

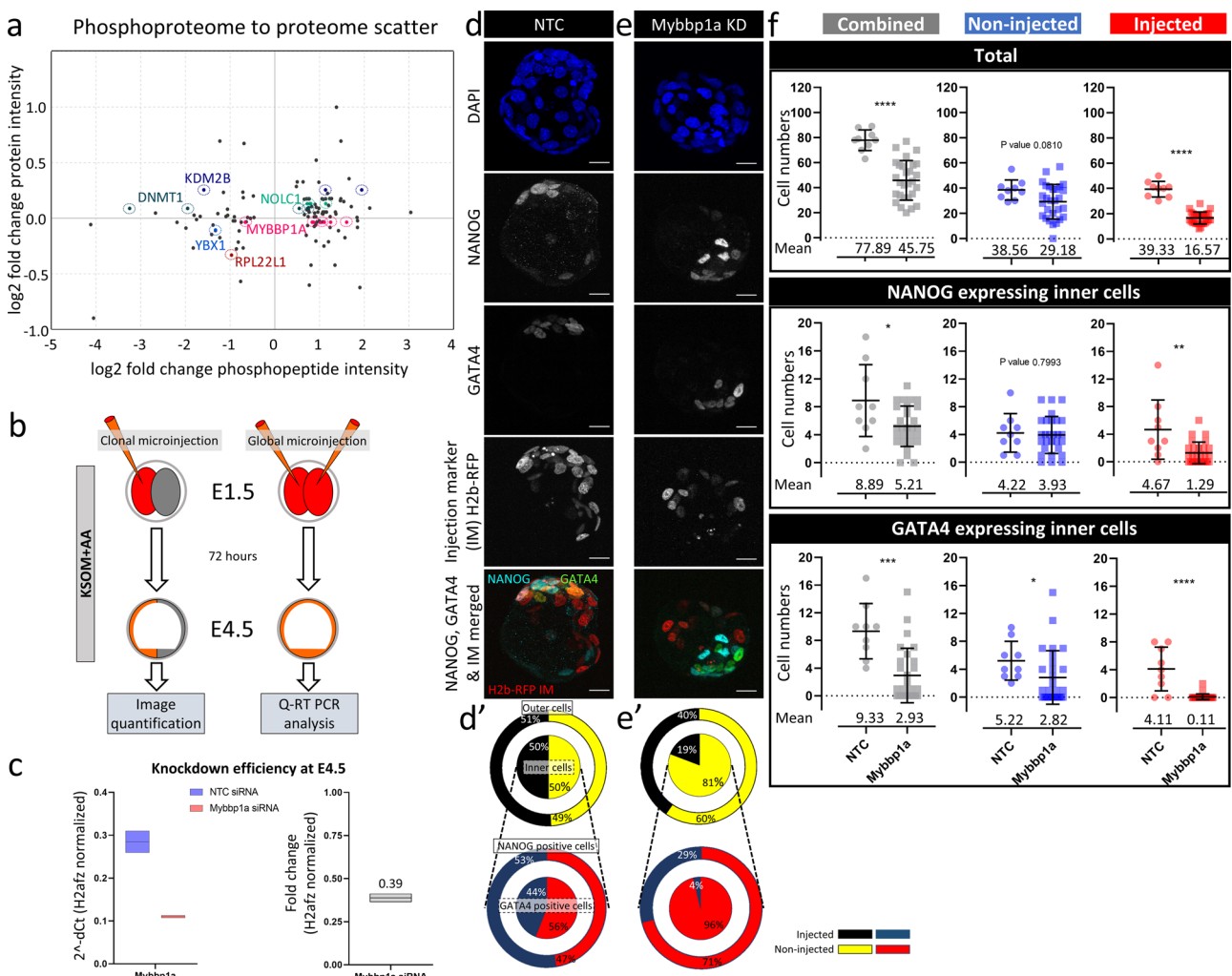

**Fig. 6 Analysis of the role of MYBBP1A (a candidate target of p38-MAPK activity) in preimplantation embryonic development and ICM cell fate specification. a** Phosphoproteome-to-proteome scatterplot for MS-detected peptides demonstrating <1.3-fold differences in abundance in the general proteome and >1.5-fold changes in phosphorylation levels. Candidates of interest (based on literature research; e.g., MYBBP1A), are highlighted (and were assayed for PrE phenotypes in clonal siRNA microinjection-mediated loss-of-function assays—**c**–**f**, Supplementary Fig. 4, and Supplementary Data 2). **b** Experimental design for analysis of the efficiency of siRNA-mediated *Mybbp1a* gene mRNA knockdown in microinjected embryos cultured to the equivalent late blastocyst (E4.5) stage (qRT-PCR analysis) and for analysis of the contributions of marked *Mybbp1a*-knockdown clones to late blastocyst cell lineages (Lineage quantification). **c** Comparison of qRT-PCR-derived relative *Mybbp1a* transcript levels (normalized to *H2afz* mRNA levels) between embryos injected with non-targeting control (NTC) siRNA and embryos injected with siRNA specific for *Mybbp1a* mRNA. *Mybbp1a* expression levels between control (blue) and *Mybbp1a* specific siRNA (red) injected embryos expressed as box plots, alongside fold-change. **d** Confocal micrograph z-projections of representative late (E4.5)-stage blastocysts initially microinjected (in one blastomere at the 2-cell stage) with NTC siRNA (*n* = 9) plus recombinant *H2b-RFP* fluorescent reporter mRNA (identifying the clonal progeny of the injected cell). Individual DAPI (blue pan-nuclear stain; indicates the total number of cells), NANOG (grayscale; EPI cells) and GATA4 (grayscale; PrE cells) channel micrographs, plus merged NANOG (cyan), GATA4 (green), and H2B-RFP (microinjected clone) images, are shown (scale bar = 20 μm). **d′** Target diagrams describing the average percentage contributions of NTC siRNA-microinjected and non-microinjected clones to either outer and inner cell populations (black and yellow targets) or mutually exclusive NANOG- and GATA4-expressing ICM cell lineages (red and blue targets). **e** Similar to **d**. **e′** is similar to **d′**, but the results were obtained after microinjection of *Mybbpa1*-specific siRNA (*n* = 28). **f** Scatterplots describing the contributions of non-microinjected (blue) and microinjected (red) cell clones (plus the combined number—gray) of NTC/*Mybbp1a* siRNA-treated embryos to the total cell count, number of inner cells expressing NANOG only, and number of inner cells expressing GATA4 only. The means and standard deviations are highlighted.

equivalent clones of control siRNA embryos), although the numbers of EPI cells were not affected (3.93 ± 2.67 versus 4.22 ± 2.77). This finding suggests that the overall reduced cell numbers of blastocysts with *Mybbp1a* clonal knockdown may also negatively affect PrE differentiation in the noninjected clones, per se (Fig. 6d–f and Supplementary Fig. 4, Supplementary Data 1).

We conclude that clonal knockdown of *Mybbp1a*, a known regulator of rRNA transcription and processing identified in this study as a candidate p38-MAPK substrate/effector during early

blastocyst maturation (Fig. 6a), biases ICM-resident clones against PrE differentiation. Hence, MYBBP1A, potentially subject to p38-MAPK activity, acts as an ICM cell fate regulator promoting PrE differentiation.

## Discussion

p38-MAPKs are well-defined stress-activated kinases involved in inflammatory diseases, including cancers, that also play

developmental roles in evolutionarily diverse species, including sea urchins, flies, frogs, zebrafish, and mice[24]. Active p38-MAPK signaling was first defined to be necessary for blastocyst formation (i.e., progression past the morula stage)[33,34]. In addition, knockout of the *Mapk14* gene is embryonic lethal and associated with defective extraembryonic placental development and adsorption of homozygous null conceptuses at E12.5[32]. Our own previous studies (confirmed here) have demonstrated that p38-MAPK activity is also required for blastocyst PrE derivation[36,37], itself a precursor of the supportive extraembryonic parietal and visceral endodermal membranes[67]. In this study, we uncovered a multilayered role for p38-MAPK-mediated translational regulation, at least partially integrated with that of the mTOR pathway and including effectors such as MYBPP1A, that coincides within a limited early blastocyst developmental window required for PrE differentiation and blastocyst cavity expansion.

The fact that p38-MAPKi blastocysts (E3.5-E4.5) exhibit cavity expansion defects that are potentially related to maturation of TE tight junctions (see below) and have impaired PrE derivation (Fig. 1)[36,37] is consistent with recent reports of positive correlations between cavity expansion and EPI and PrE cell fate specification and sorting[13]. For example, Ryan and colleagues found that both chemical modulation (using ouabain, an inhibitor of the ATP1 channel responsible for cavity fluid accumulation[68–70]) and mechanical modulation of blastocyst cavity size during early blastocyst maturation (E3.5-E4.0) reduces PrE marker protein expression (assayed at E4.0)[13]. Strikingly, the p38-MAPKi-associated PrE differentiation defects we have reported (here and previously[36,37]) closely resemble those of mechanically deflated blastocysts[13] (reduced GATA4 expression and normal EPI specification as indicated by either NANOG or SOX2 expression analysis). Indeed, late blastocyst p38-MAPKi PrE phenotypes involve a statistically significant lack of GATA4-positive ICM cells compensated for by a population of unspecified progenitors coexpressing NANOG and GATA6[36,37]. We hypothesize that a similar population of unspecified PrE progenitors may also be found in mechanically deflated blastocysts[13]. Thus, it is possible that the impaired blastocyst PrE differentiation p38-MAPKi phenotypes are, at least in part, mediated by TE defects resulting in insufficient cavity expansion. In fact, p38-MAPKi has previously been reported to increase TE tight junction permeability and reduce *Aqp3* expression[34], which together with Na$^+$/K$^+$ pumps (such as ATP1) initiate and maintain blastocyst cavity expansion and luminal pressure[34,71] (reported to more than double between E4.0 and E4.5[71]). Interestingly, our (phospho-) proteomic and transcriptomic data (derived from both TE and ICM lineages) indicate that p38-MAPKi induced downregulation of *Tjp1* and *Atp1a1* (respectively encoding tight junction protein 1 and one of the two subunits of ATP1—deletion of which causes peri-implantation lethality and cavity collapse after in vitro culture[72]) at the protein and RNA levels (~50 and 40%, respectively—Supplementary Data 1 tabs marked Figs. 2c, 3b (+7 h) and Supplementary Table 1). Whether this reflects targeted regulation of such specific genes or is related to more general p38-MAPK mediated control of protein translation is not clear. Importantly, PrE differentiation can still occur in the absence of TE or a blastocyst cavity. For example, ICM immunosurgically isolated from E3.5-stage mouse blastocysts can support the formation of an outer layer of PrE within 24 h (mirroring PrE differentiation in intact blastocysts)[73], as can mouse embryonic stem (ES) cell-derived ICM organoids[74].

Within the blastocyst ICM, the role of FGF4-FGFR1/2-ERK1/2 signaling in appropriate lineage specification is well established[4,5,9,10]. ERK pathway inhibition (targeting MEK1/2) throughout the blastocyst maturation period completely blocks PrE differentiation; however, removal of MEK1/2 inhibition at

any point prior to ~E4.25 is still permissive for PrE differentiation[9,10,36,75]. These data indicate the persistence of FGF4-based signaling from emerging EPI cells that, under unperturbed developmental conditions, drive the incremental specification and differentiation of PrE as blastocysts mature (ensuring a stable EPI:PrE ratio)[9]. The extent to which active FGF4 signaling may impact the PrE-supportive role of p38-MAPK in the mouse blastocyst ICM is questionable. Stimulation of p38-MAPK activity by various exogenous FGF ligands has been reported to occur in various cell models of proliferation and differentiation, although it has not been thoroughly characterized[76–80], and a role for FGF2 (acting via FGFR2) requiring p38-MAPK in mouse blastocyst TE formation has been described[81]. In the E3.75-stage mouse blastocyst ICM, *Fgfr1* is equally expressed in all cells irrespective of *Fgf4* expression, whereas *Fgfr2* is expressed at 4- to 6-fold higher levels in cells not expressing *Fgf4* (presumptive PrE progenitors)[82]. However, *Fgfr1* is the primary receptor directing PrE formation[4,5]. Interestingly, we observed reduced mRNA expression of *Fgfr1* and *Fgfr2* that was strongest at the E3.5 + 7 h time-point, after p38-MAPKi, in our mRNA-Seq dataset (Supplementary Data 1 see tab marked Fig. 3b (+7 h)). However, *Fgf4* and double-*Fgfr1/2* gene knockouts have phenotypes that developmentally precede the newly identified p38-MAPKi sensitive window (relating to PrE differentiation). For example, they are associated with complete loss of GATA6 protein expression through the early blastocyst stages and result in all ICM cells solely expressing EPI markers (although GATA6 expression is normally initiated together with NANOG in all nascent ICM cells)[4,5,7]. This situation contrasts with p38-MAPKi phenotypes in which NANOG and GATA6 coexpression persist at the expense of PrE, but not EPI, differentiation[36,37]. Moreover, ERK phosphorylation, indicative of activated FGF4 signaling, begins at approximately E3.0-E3.25 and increases with time in a manner correlated with increasing GATA6 expression[82]. Additionally, we found that the reported pan-ICM conversion of cells to a PrE fate caused by exposure to exogenous FGF4[5,57,83,84] is not sensitive to concomitant p38-MAPKi, unlike in the case of MEK1/2 inhibition (Supplementary Fig. 5 and 6 and Supplementary Data 1), although we acknowledge concerns about potentially supraphysiological FGF4 concentrations[85]. Collectively, these data indicate that the PrE-promoting role of p38-MAPK most likely does not function directly downstream of classically activated FGF4-FGFR1/2-ERK1/2 signaling or is at least not limited to this pathway. Non-MEK1/2-dependent activation of p38-MAPKs has been reported in multiple human *FGFR2* genetic mutation-induced pathophysiologies[26,27], suggesting the possible involvement of noncanonical FGFR2-mediated p38-MAPK activation mechanisms (as reviewed previously[25]). Considering the proposed PrE survival-specific role of FGFR2[16] in association with PDGFRa[14–16] and the p38-MAPKi-mediated PrE phenotypes we have described, it is tempting to speculate that such noncanonical signaling cascades may contribute (together with classical ERK pathway activation) to mouse blastocyst ICM cell fate derivation, albeit at slightly later developmental time points than the classical cascade. Another potential mechanism of emerging PrE-specific p38-MAPK activation may be via activated FGFR3. Although they are expressed at equally low levels in all early blastocyst (E3.5) ICM cells, *Fgfr3* transcripts are specifically enriched in an *Fgf4*-dependent manner in late blastocyst (E4.5) PrE[31]. Moreover, FGFR3 has been reported to activate p38-MAPK[30]. The PrE specificity of PDGFRa is well documented, and fluorescently tagged genetic knock-ins and knockouts are regularly used to identify PrE cells[13,15,16,86]. A recent report described PDGFRa-mediated signaling via PI3K and mTOR as enabling PrE survival[15]. Although inhibition of JAK or mTORC1 (one of two

mTOR-containing complexes—see below) alone does not impact any preimplantation cell lineages, combined inhibition is particularly detrimental to PrE survival. Our observations that p38-MAPKi PrE phenotypes can be circumvented (at least partially) by mTOR activation implies that p38-MAPK acts functionally upstream of mTOR and downstream of potentially commonly activated receptor kinases, such as FGFR2, FGFR3, PDGFRa, or others.

Notwithstanding the above discussion points regarding p38-MAPK activation, the central finding that p38-MAPK plays a fundamental role in regulating protein translation during early blastocyst maturation is consistent with the creation of metabolically permissive states compatible with PrE specification and differentiation. The finding that p38-MAPKi PrE deficits can be partially reversed by concomitant mTOR activation (Fig. 5f, j–q) is important and emphasizes the importance of p38-MAPK-mediated regulation of translation. The mTOR signaling pathway (integrated by two mTOR-containing complexes, mTORC1 and mTORC2) is a global regulator of translation and maintains cellular energy homeostasis in response to extracellular stimuli such as growth factors and nutrient availability[51]. As part of mTORC1, mTOR specifically controls transcription and translation of ribosomal genes/proteins, synthesis and processing of rRNA, and biogenesis and availability of ribosomes to ultimately dictate general translation of all mRNA transcripts (mTORC2 is associated with cotranslational protein degradation and lipolysis)[51,87–90]. Our data indicate that there is considerable overlap between the described mTOR function and the role of p38-MAPK during early blastocyst maturation and eventual PrE formation. However, despite the similarities and the partial reversal of the p38-MAPKi-induced PrE differentiation phenotype caused by simultaneous activation of mTOR, it is clear that p38-MAPKi does not merely phenocopy mTOR inhibition-induced diapause[54]. Rather, the p38-MAPKi-induced phenotype is specifically characterized by impaired PrE differentiation in the context of ongoing development (EPI specification is not impaired), which nevertheless has all the hallmarks of impaired translation. Notably, the restoration of GATA4-positive PrE cells in blastocysts subjected to both p38-MAPKi and mTOR activator treatment was frequently associated with coexpression of NANOG (Fig. 5f, q), suggesting the occurrence of a pseudo-state that does not normally exist in which PrE differentiation was uncoupled from the ordinarily required need to downregulate NANOG. The PrE-specific effect of p38-MAPKi was further illustrated by the finding that the number of EPI cells (defined by sole expression of NANOG) was not altered regardless of cotreatment with the mTOR activator (Fig. 5m). We also noted that the partial PrE differentiation "rescue" effects of mTOR activation on p38-MAPKi blastocysts were not associated with enhanced blastocyst cavity expansion (Supplementary Fig. 7), suggesting that these and the originally observed p38-MAPKi phenotypes are at least partly PrE progenitor cell autonomous.

The observed p38-MAPKi-induced upregulation of ribosomal protein transcripts (peaking at E3.5 + 7 h—Fig. 3; also associated with downregulation at the protein level—Figs. 2c and 3e) suggested accumulation caused by confirmed overall impaired translation (Fig. 4b–d) that most likely involved rRNA processing defects (Fig. 4e, f). A previous study has described HTATSF1 as a regulator of rRNA/mRNA transcriptional and post-transcriptional processing, protein synthesis, and the pre- to postimplantation transition from the embryonic naïve to primed pluripotent states[46]. HTATSF1-mediated repression of active translation during this transitory period is associated with increased intron retention in ribosomal protein-encoding mRNAs, which causes degradation of the mRNAs via the nonsense-mediated decay pathway[46]. Although we also observed reduced translation after p38-MAPKi (Figs. 2c and 4b–d), we did not observe any similarly enhanced intron retention in our mRNA-Seq dataset (Supplementary Table 2). However, MYBPP1A, a confirmed HTATSF1 interaction partner[46], was identified in our phosphoproteomic screen for candidate p38-MAPK effectors (Fig. 6a), clonal dysregulation of which was associated with PrE deficits (Fig. 6d–f). Other cell culture studies have implicated MYBBP1A in the transcriptional repression of rRNA genes and in the facilitation of pre-rRNA processing[61]. We propose that MYBBP1A (under the regulation of p38-MAPK) may act as a checkpoint agent enabling switching from homeostasis of cellular protein synthesis and the coordination of a functionally important translational response to externally provided differentiation cues (via regulation of mature rRNA levels available for functional ribosome biogenesis), as supported by reports in other models describing MYBBP1A-regulated rRNA expression and processing in the cellular stress response[63,64]. Interestingly, MYBBP1A is known to suppress mitochondrial respiration, and we identified mitochondrial respiration-related transcripts as the second most statistically significant ontological class of gene transcripts with p38-MAPKi-mediated differential expression (Fig. 3d—representing upregulated mRNA levels). Our phosphoproteomic screening data revealed multiple p38-MAPKi-sensitive phosphorylation sites within the MYBBP1A protein (Fig. 6a and Supplementary Data 1); phosphorylation at serine 1280 (S1280) was decreased, but phosphorylation at S1303 and S1323, among other sites, was increased. Our results indicate that S1280 is a good candidate direct substrate for p38-MAPK, particularly as it contains the [S/T]P consensus phosphorylation motif for proline-directed mitogen-activated kinases in general[44] (Supplementary Data 2), but this remains to be experimentally validated. To our knowledge, the only verified MYBBP1A phosphorylation site is centered on S1303 and is a reported Aurora B kinase (AURKB) substrate[91]. The mechanistic details of how such phosphorylation events functionally impact MYBBP1A and to what extent they are involved in the described early blastocyst p38-MAPKi-induced translation and PrE phenotype remain to be examined.

In summary, we describe a role for p38-MAPK activity during a limited developmental window of early blastocyst maturation that is centered on the regulation of translation. This role ensures germane ICM cell fate specification, specifically facilitating PrE derivation by the late blastocyst (E4.5) stage. We propose that such positive translational regulation is at least partially integrated with the mTOR pathway and is necessary to prime subsequent differentiation of PrE progenitors (see model—Fig. 7).

## Methods

**Mouse lines and embryo culture**. All mouse-related experimental procedures (i.e., collection of preimplantation-stage embryos for further study) complied with the Animal Research: Reporting of In Vivo Experiments (ARRIVE) guidelines and were carried out in accordance with EU directive 2010/63/EU (for animal experiments). The superovulation and strain mating regimens used to produce experimental embryos were as follows: F1 hybrid (C57Bl6 × CBA/W) females were intraperitoneally injected with 7.5 IU of PMSG (Folligon, MSD Animal Health, Boxmeer, Netherlands) and again with 7.5 IU of hCG (Sigma-Aldrich, St. Louis, Missouri, USA; Cat. No. CG10) after 48 h and then mated overnight with F1 males (successful mating was confirmed the next morning by the presence of vaginal plugs). E1.5 (2-cell)-stage embryos were isolated from female oviducts in M2 medium (prewarmed at 37 °C for at least 2–3 h) and thereafter cultured in KSOM (Sigma-Aldrich, EmbryoMax KSOM Mouse Embryo Media; Cat. No. MR-020P-5F—prewarmed and equilibrated in 5% CO$_2$ and 37 °C) either with or without AA supplementation. For KSOM + AA conditions, Gibco MEM Non-Essential Amino Acids Solution (100×) (ThermoFisher Scientific, Paisley, Scotland; Cat. No. 11140035) and Gibco MEM Amino Acids Solution (50×) (ThermoFisher Scientific; Cat. No. 11130036) were used at working concentrations of 0.5×. Embryos were cultured in microdrops prepared in 35 mm tissue culture dishes covered with light mineral oil (Irvine Scientific, Santa Ana, California, USA; Cat. No. 9305) in 5% CO$_2$ incubators maintained at 37 °C until the appropriate stage. Pharmacological manipulations were carried out via the addition of chemical agents dissolved in DMSO (Sigma-Aldrich; Cat. No. D4540) to KSOM ± AAs (the concentration details,

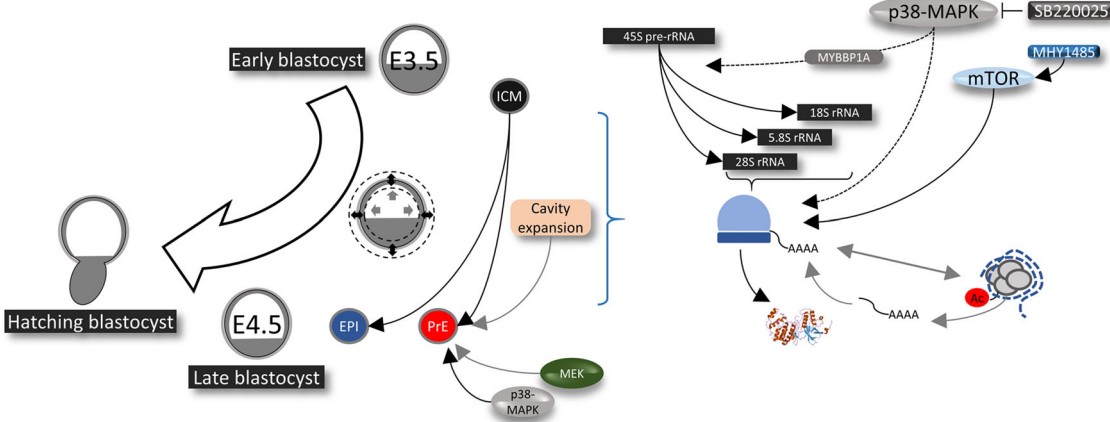

**Fig. 7 Working model of the role of p38-MAPK in regulating protein translation to prime PrE differentiation during preimplantation-stage blastocyst maturation.** During the period of blastocyst maturation (E3.5 to E4.5), p38-MAPKs, potentially acting via MYBBP1A, enable pre-rRNA processing and regulates global translational and co-transcriptional events, which is necessary for normal development and PrE specification. p38-MAPKs appear to be acting upstream of mTOR, towards a pathway supporting PrE survival in the maturing blastocyst.

durations, etc. are given in Supplementary Table 3). Equivalent volumes of solvent control were used at final working concentrations of 0.2–0.5% by volume. The concentrations of the p38-MAPK inhibitor (SB220025—20 μM) and MEK1/2 inhibitor (PD0325901—1 μM) used were derived from empirical titrations that we performed previously[36] and/or from published literature[10,33]. All KSOM-based culture media with or without additional chemicals (AAs, pharmacological agents) were prewarmed and equilibrated in 5% $CO_2$ and 37 °C for at least 3–4 h prior to embryo transfer.

**Sample collection for mass spectrometric analysis of the differential (phospho-)proteome.** Two-cell (E1.5)-stage embryos were cultured in normal KSOM + AA conditions until E3.5 + 2 h. Thereafter, 300 embryos each were moved to control or p38-MAPKi conditions and cultured for another 7 h (E3.5 + 9 h). The embryos were then washed with prewarmed Hank's balanced salt solution (HBSS, Sigma-Aldrich; Cat. No. H9269) and lysed in 1.5 ml centrifuge tubes containing ~15 μl of SDT lysis buffer (4% (w/v) SDS, 100 mM Tris-HCl pH 7.6, 0.1 M DTT) in a 95 °C heat block for 12 min. The tubes were briefly centrifuged at 750 rpm, cooled to room temperature, and stored at −80 °C.

**Sample preparation for liquid chromatography–mass spectrometry (LC–MS) analyses.** Individual protein solutions were processed by the filter-aided sample preparation (FASP) method[92] with some modifications. The samples were mixed with 8 M UA buffer (8 M urea in 100 mM Tris-HCl, pH 8.5), loaded onto a Microcon device with a molecular weight cutoff (MWCO) of 30 kDa (Merck Millipore, Burlington, Massachusetts, USA) and centrifuged at 7000 × g for 30 min at 20 °C. The retained proteins were washed (all centrifugation steps after sample loading were performed at 14,000 × g) with 200 μl of UA buffer. The final protein concentrates retained in the Microcon device were mixed with 100 μl of UA buffer containing 50 mM iodoacetamide and incubated in the dark for 20 min. After the next centrifugation step, the samples were washed three times with 100 μl of UA buffer and three times with 100 μl of 50 mM NaHCO₃. Trypsin (1 μg, sequencing grade, Promega, Madison, Wisconsin, USA) was added onto the filter, and the mixture was incubated for 18 h at 37 °C. The tryptic peptides were finally eluted by centrifugation followed by two additional elutions with 50 μl of 50 mM NaHCO₃. The peptides were then cleaned via liquid-liquid extraction (3 iterations) using water-saturated ethyl acetate[93].

One-tenth of the cleaned FASP eluate was removed for direct LC–MS measurements and evaporated completely in a SpeedVac concentrator (ThermoFisher Scientific, Waltham, Massachusetts, USA). The peptides were further transferred into LC–MS vials using 50 μl of 2.5% formic acid (FA) in 50% acetonitrile (ACN) and 100 μl of pure ACN with added polyethylene glycol (final concentration 0.001%) and concentrated in a SpeedVac concentrator.

Phosphopeptides were enriched from the remaining 9/10th of the cleaned FASP eluate after complete solvent evaporation (in a SpeedVac concentrator) using a High-Select TiO₂ Phosphopeptide Enrichment Kit (ThermoFisher Scientific) according to the manufacturer's protocol. Phosphopeptide standards (0.1 pmol of MS PhosphoMix 1, 2, 3 Light; Sigma-Aldrich) were added to the suspended sample in binding/equilibration buffer. The flow-through fraction was dried and used for the second enrichment step using a High-Select Fe-NTA Phosphopeptide Enrichment Kit (ThermoFisher Scientific) according to the manufacturer's protocol. The resulting phosphopeptides were extracted into LC–MS vials with 2.5% FA in 50% ACN and 100% ACN with added polyethylene glycol (final concentration 0.001%) and concentrated in a SpeedVac concentrator.

**LC–MS analysis of peptides.** LC–MS/MS analyses were performed for all peptide mixtures (3 peptide solutions for each sample: (1) not enriched, (2) enriched with phosphopeptides using a TiO₂ enrichment kit, and (3) enriched with phospho-peptides using an Fe-NTA enrichment kit) using an RSLCnano system connected to an Orbitrap Fusion Lumos mass spectrometer (ThermoFisher Scientific). Prior to LC separation, the tryptic digests were concentrated online and desalted using a trapping column (100 μm × 30 mm, column compartment temperature of 40 °C) filled with 3.5 μm X-Bridge BEH 130 C18 sorbent (Waters). After washing the trapping column with 0.1% FA, the peptides were eluted (flow rate: 300 nl/min) from the trapping column onto an analytical column (Acclaim PepMap 100 C18, 3 μm particles, 75 μm × 500 mm, column compartment temperature of 40 °C, ThermoFisher Scientific) by using a 50- or 100-min nonlinear gradient program (1–56% mobile phase B; mobile phase A: 0.1% FA in water; mobile phase B: 0.1% FA in 80% ACN) for analysis of phosphopeptide-enriched fractions or non-enriched peptide mixtures, respectively. The trapping column and the analytical column were equilibrated prior to sample injection into the sample loop. The analytical column outlet was directly connected to a Digital PicoView 550 (New Objective) ion source with the sheath gas option and SilicaTip emitter (New Objective; FS360-20-15-N-20-C12) utilization. An Active Background Ion Reduction Device (ABIRD, ESI Source Solutions, Woburn, Massachusetts, USA) was installed.

MS data were acquired in a data-dependent strategy with a cycle time of 3 s and with a survey scan (350–2000 m/z). The resolution of the survey scan was 60,000 (200 m/z) with a target value of $4 × 10^5$ ions and a maximum injection time of 50 ms. Higher-energy collisional dissociation (HCD) MS/MS (30% relative fragmentation energy, normal mass range) spectra were acquired with a target value of $5.0 × 10^4$. The MS/MS spectra were recorded in Orbitrap at resolving power of 30,000 or 15,000 (200 m/z), and the maximum injection time for MS/MS was 500 or 22 ms for analysis of phosphopeptide-enriched fractions or non-enriched peptide mixtures, respectively. Dynamic exclusion was enabled for 60 s after one MS/MS spectrum acquisition. The isolation window for MS/MS fragmentation was set to 1.6 m/z.

The raw mass spectrometric data files were analyzed using MaxQuant software (version 1.6.6.0) with the default settings unless otherwise noted. MS/MS ion searches were performed against the modified Common Repository of Adventitious Proteins (cRAP) database (based on http://www.thegpm.org/crap, 112 protein sequences) containing protein contaminants such as keratin and trypsin and the UniProtKB protein database for *Mus musculus* (ftp://ftp.uniprot.org/pub/databases/uniprot/current_release/knowledgebase/reference_proteomes/Eukaryota/UP000000589_10090.fasta.gz; downloaded 8.5.2019, version 2019/05, number of protein sequences: 22287). Oxidation of methionine and proline, deamidation (N, Q), and acetylation (protein N terminus) were the optional modifications, and the trypsin/P enzyme with 2 allowed missed cleavages and a minimum peptide length of 6 AAs were set. Only peptides and proteins meeting a false discovery rate (FDR) threshold <0.01 and proteins with at least one unique or razor peptide were considered. Match-between-runs was set separately for all enriched and non-enriched peptide solution analyses. The same experimental name was used for differently enriched peptide solution analyses coming from the same sample.

The protein intensities reported in the proteinGroups.txt file and the evidence intensities reported in the evidence.txt file (output from the MaxQuant program) were further processed using the OmicsWorkflows software container environment (https://github.com/OmicsWorkflows), version 3.7.1a. The processing workflow is available upon request. Briefly, the workflow featured (1) protein level data processing, including (a) removal of decoy hits and contaminant protein groups,

(b) log2 transformation of protein group intensities, (c) LoessF normalization and (d) differential expression using the LIMMA statistical test (qualitative changes were considered separately without statistical evaluation); (2) phosphopeptide level data processing, including (a) removal of evidence associated with non-phosphorylated and contaminant proteins, (b) grouping of the evidence for the identical combinations of peptide sequences and sets of modifications, (c) LoessF normalization and (d) differential expression analysis using the LIMMA statistical test (qualitative changes were considered separately without statistical evaluation). Differentially abundant phosphopeptide candidates (phosphopeptides exhibiting ≥1.5-fold differential abundance; note that the cutoff was applied to acknowledge the potential sensitivity threshold limitations when such scarce starting material is used) were selected based on the criteria mentioned within the Results section. The mass spectrometry proteomics data have been deposited to the ProteomeXchange Consortium via the PRIDE[94] partner repository with the dataset identifier PXD025711.

### mRNA sequencing (mRNA-Seq) library preparation and sequencing.
Two-cell (E1.5)-stage embryos were cultured in normal KSOM + AA conditions until E3.5. Thereafter, 30 embryos each were moved to control or p38-MAPKi conditions and cultured for another 4, 7, or 10 h. The embryos were then processed for total RNA isolation using the ARCTURUS PicoPure RNA Isolation Kit (ThermoFisher Scientific; Cat. No. KIT0204) following the manufacturer's protocol. Oligo dT-based mRNA isolation was carried out using an NEBNext Poly(A) mRNA Magnetic Isolation Module (NEB, Ipswich, Massachusetts, USA; Cat. No. E7490), and libraries were prepared using an NEBNext Ultra II Directional RNA Library Prep Kit for Illumina (NEB; Cat. No. E7760), as directed by the manufacturer. The libraries were sequenced as 50 bp single-end reads on an Illumina HiSeq 4000 platform. The mRNA-Seq datasets are available in the Gene Expression Omnibus (GEO) database with accession number GSE162233.

### Data processing and bioinformatics analysis.
The sequenced reads were adapter- and quality-trimmed using Trim Galore! v0.4.1 and mapped to the mouse GRCm38 genome with HISAT2 v2.0.5. Differentially expressed genes between control and p38-MAPKi embryos at individual time points were identified by DESeq2 within SeqMonk v1.45.4 using GRCm38_v90 transcriptome annotation. Hierarchical clustering was performed in SeqMonk v1.45.4 with mean-centered expression values. Gene Ontology (GO) analysis was performed using GOrilla[95,96]. Due to the omission of spiked-in controls during mRNA-Seq library preparation, it was not possible to assay normalized global transcriptional output from these data.

### Embryo manipulation via microinjection.
Single (for immunofluorescence confocal microscopy) or double (for qRT-PCR) blastomere microinjections of 2-cell (E1.5)-stage embryos were performed using a FemtoJet or FemtoJet 4i (Eppendorf, Hamburg, Germany; Cat. No. 5252000013) microinjector, a mechanical micro-manipulator (Leica, Wetzlar, Germany; Cat. No. ST0036714) and a CellTram Vario (Eppendorf; Cat. No. 5176000033) pneumatic handler under a negative capacitance-enabled current controlled by an Electro 705 Electrometer (WPI, Sarasota, Florida, USA; Cat. No. SYS-705) and on the stage of an Olympus IX71 inverted fluorescence microscope. The embryos were pneumatically handled and immobilized for microinjection using a borosilicate glass capillary holder (without filament—Harvard Apparatus, Holliston, Massachusetts, USA; Cat. No. 30-0017). Injection needles prepared from filamented borosilicate glass capillaries (Harvard Apparatus; Cat. No. 30-0038) using a Narishige PC-10 capillary glass needle puller (Narishige Scientific Instrument Lab., Tokyo, Japan) were connected to the microinjector. All siRNAs (Supplementary Table 4) were co-microinjected at 10 μM concentrations with 50 ng/μl *H2b-RFP* mRNA in prewarmed drops of M2 medium overlaid with mineral oil on the surfaces of concave microscope slides.

### Immunofluorescence staining, confocal microscopy, and image acquisition.
To remove the *zona pellucida*, blastocysts were quickly washed and pipetted in pre-warmed drops of acidic Tyrode's solution (Sigma-Aldrich; Cat. No. T1788) until the *zona pellucida* was visually undetectable and were then immediately washed in prewarmed drops of M2 medium. Thereafter, the embryos were fixed in the dark with 4% paraformaldehyde (Santa Cruz Biotechnology, Inc., Dallas, Texas, USA; Cat. No. sc-281692) for 20 min at room temperature. Permeabilization was performed by transferring the embryos to a 0.5% solution of Triton X-100 (Sigma-Aldrich; Cat. No. T8787) in phosphate-buffered saline (PBS) for 20 min at room temperature. All washes after the fixation, permeabilization, and antibody staining procedures were performed in PBS with 0.05% TWEEN 20 (Sigma-Aldrich; Cat. No. P9416) (PBST) by transferring the embryos between two drops of PBST or wells (of 96-well microtiter plates) containing PBST for 20 min at room temperature. Blocking and antibody staining was performed in 3% bovine serum albumin (BSA; Sigma-Aldrich; Cat. No. A7906) in PBST. For blocking, embryos were incubated for 30 min at 4 °C before both primary and secondary antibody staining. For primary antibody staining (in blocking buffer), embryos were incubated overnight (~16 h) at 4 °C. Secondary antibody staining was carried out in the dark at room temperature for 70 min. The stained embryos were mounted with VECTASHIELD mounting medium containing DAPI (Vector Laboratories, Inc., Burlingame, California, USA; Cat. No. H-1200), placed under coverslips on glass-

bottomed 35 mm culture plates, and incubated at 4 °C for 30 min in the dark prior to confocal imaging. Details of the primary and secondary antibody combinations used can be found in Supplementary Table 5. Confocal images were acquired using an FV10i confocal laser scanning microscope and FV10i-SW image acquisition software (Olympus, Tokyo, Japan). The images were analyzed using FV10-ASW 4.2 Viewer (Olympus) and Imaris X64 Microscopy Image Analysis Software (version 6.2.1; Bitplane AG—Oxford Instruments plc, Abingdon, United Kingdom). The cells were counted both manually and semi-automatically using Imaris X64.

### Time-lapse imaging.
Embryos at E3.5 were placed in 20 μl droplets of KSOM + AAs (+DMSO or +SB220025) in 16-well dishes (PrimoVision, Vitrolife, Gothenburg, Sweden) at a density of one embryo per well within a PrimoVision imaging system and cultured for 24 h (until E4.5) under standard culture conditions. The embryos were imaged every 10 min, the equatorial areas were recorded, and the data were analyzed (Fig. 1e and as described previously[97]).

### Scarce-sample polysome (SSP) profiling.
Two-cell (E1.5)-stage embryos were cultured in KSOM + AAs and transferred to control or p38-MAPKi conditions for 10 h starting from E3.5 (E3.5 + 10 h; as depicted in Fig. 4a). Prior to harvesting, the embryos were transferred to cycloheximide (0.1 mg/ml)-containing medium for 10 min and then washed 3× in PBS (supplemented with 0.1 mg/ml cycloheximide) and 1× in transfer medium (0.1% PVA in PBS + 0.1 mg/ml cyclohexamide). The embryos were then transferred to 0.5 ml low DNA-binding tubes in a minimal volume of a transfer medium, flash-frozen in liquid nitrogen, and stored at −80 °C. Cellular lysis and sucrose gradient centrifugation to procure ribosomal fractions followed by isolation of corresponding rRNA were performed as described previously[45]. The resultant fractions representing rRNA corresponding to non-polysomes (fractions 1–5; from the upper (lighter) fractions) and polysomes (fractions 6 to 10; from the lower (heavier) fractions) were resuspended in 20 μl of nuclease-free water. cDNA was then prepared. Briefly, 4.0 μl of total RNA was mixed with 1.6 μl of 0.183 μg/μl random hexamers and 3.0 μl of nuclease-free water. The samples were incubated on a heat block at 70 °C for 5 min, placed on ice for 5 min and then briefly microfuged. Then, 3.9 μl of nuclease-free water, 4.0 μl of 5× reaction buffer, 1.0 μl of RevertAid H Minus reverse transcriptase (Thermo-Fisher Scientific; Cat. No. EP0451), 0.5 μl of RiboLock RNase inhibitor (Thermo-Fisher Scientific; Cat. No. EO0381) and 2.0 μl of dNTP mix (ThermoFisher Scientific; Cat. No. R0192) were added. The preparations were processed in a thermocycler with the following program: 25 °C for 10 min, 37 °C for 5 min, 42 °C for 75 min and 70 °C for 10 min. The synthesized cDNA was stored at −20 °C, and 2.0 μl aliquots were used as templates for qRT-PCR (as described previously[45]).

### O-propargyl-puromycin (OPP) quantification of de novo protein synthesis.
Mouse embryos were cultured in vitro from E1.5 to E3.5 in KSOM + AAs and then further cultured under control or p38-MAPKi conditions for 10 h (E3.5 + 10 h). OPP staining was then performed using a Click-iT Plus OPP Alexa Fluor 488 Protein Synthesis Assay Kit (ThermoFisher Scientific; Cat. No. C10456) following the prescribed protocol with modifications for in vitro preimplantation embryo culture. In brief, equilibrated culture dishes (described above) containing drops of KSOM (without AA supplementation) and 40 μM solutions of OPP supplemented with either control DMSO or 20 μM SB220025 were prepared. Following a 60-min pre-incubation period, control and p38-MAPKi blastocysts were transferred to their respective OPP-containing dishes and incubated for an additional 60 min. Thereafter, the embryos were fixed, permeabilized, and washed as described above (immunofluorescence confocal microscopy). To enable fluorescent visualization of the incorporation of OPP into nascent polypeptides, Click-iT reactions were set up as described in the kit manual (with Click-iT reaction buffer, copper protectant, Alexa Fluor 488 picolyl azide, and reaction buffer additive). The fixed embryos were incubated in the reaction mixture for 25 min at room temperature in the dark. Thereafter, the embryos were washed in a 1:1 mixture of kit-provided wash buffer and PBST, mounted in VECTASHIELD, imaged using standard confocal microscope settings (as described above) to detect the Alexa Fluor 488 fluorophore and quantified as described.

### Cell number quantification, statistical analysis, and graphical representation.
The cells counted as part of the total embryo cell numbers (plus outer and inner cell populations) from confocal acquired micrographs (based on DAPI nuclear staining) were further subcategorized as EPI or PrE cells based on detectable and exclusive NANOG and GATA4 protein expression (confocal images in Figs. 1b, 5 and 6 and graphs in Fig. 1b, c, 5 and 6). Cells not located within blastocyst ICMs that also did not stain for GATA4 and/or NANOG were designated outer/TE cells. In relation to Fig. 5p specifically, ICM cells that were positively stained for both GATA6 and NANOG at E4.5 were designated as uncommitted/unspecified in terms of cell fate, and the population of similarly GATA4- and NANOG-positive ICM cells (primarily resulting from coinhibition and coactivation of p38-MAPK and mTOR, respectively—Fig. 5f) was also recorded. In Fig. 4g, h, cells were designated "outer cells" based on detectable CDX2 (TE marker[48]) expression. The relative levels of H4K16ac-modified chromatin (Fig. 4g) were quantified to determine global chromatin status, and RNA pol II S2p (Fig. 4h) levels were quantified to determine active transcriptional elongation, as described below in the section on

fluorescence intensity quantification and statistical analysis. Initial recording and data accumulation were carried out using Microsoft Excel, and further statistical analysis and graphical representations were performed with GraphPad Prism 8. Based on the normality and lognormality comparisons, appropriate statistical tests were used for the compared datasets (summarized in Supplementary Data 2). Unless otherwise stated within individual graphs with specific $P$ values (if the differences were statistically nonsignificant), the following significance intervals were recognized: $P < 0.0001$ (****), 0.0001–0.001 (***), 0.001–0.01 (**), and 0.01–0.05 (*). All dot plots represent the total sample size, with the associated means and standard deviation bars provided.

**Fluorescence intensity quantification and statistical analysis**. The fluorescence intensity pertaining to OPP incorporation (Fig. 4b) was quantified for the minimum complete confocal $z$-series of each embryo as a whole using Fiji (ImageJ)[98]. The confocal images from the channel pertaining to the 488 nm wavelength were analyzed as follows: Image > Stacks > Z Project. The projection-type "Sum Slices" was used to obtain a $Z$-projected image for the entire embryo. The levels of H4K16ac (Fig. 4g) and RNA pol II S2p (Fig. 4h) were quantified as the fluorescence intensity per cell nuclei, and the cells were defined as inner or outer cells based on the absence or presence of CDX2 immunofluorescence, respectively. The measurements for all of the above were set with the command Analyze > Set Measurements, and the following options were chosen: Area, Mean gray value, and Integrated density. Using the "Polygon selections" tool, an area encompassing the $Z$-projected embryo image (for OPP) or individual cell nuclei (for H4K16ac and RNA pol II S2p) was demarcated, and the aforementioned measurements were recorded. The selected area was then moved to encompass an area excluding the embryo or cell nucleus, and background measurements were recorded. This process was continued for all the embryos analyzed under both control and p38-MAPKi conditions, and the results were transferred to a spreadsheet for further calculations. The corrected total cell fluorescence (CTCF), in arbitrary units, was measured for each embryo as follows: CTCF = integrated density – (area of selected cell × mean fluorescence of background readings)[99,100] (Supplementary Data 1 tabs marked Fig. 4b, g, h). The calculated CTCFs are plotted in scatterplots, with the means and standard deviations marked. The CTCF differences were statistically tested using the Mann–Whitney test, and the results, unless otherwise stated within individual graphs with specific $P$ values (if statistically non-significant), are marked according to the following significance intervals: $P < 0.0001$ (****), 0.0001–0.001 (***), 0.001–0.01 (**) and 0.01–0.05 (*).

**Blastocyst size and volume calculations**. The equatorial areas of blastocoel cavities (Fig. 1d) were calculated by measuring the inner circumference of the centrally located widest $Z$-stack using Fiji (ImageJ)[97]. To compare the volumes of p38-MAPK-inhibited blastocoel cavities with those reported by Ryan et al.[13], we deduced the volumes from inner circumference measurements. The calculated areas and corresponding volume measurements are tabulated in Supplementary Data 1 (tab marked Fig. 1d). The equatorial area and total volume measurements for whole blastocysts were obtained by similarly measuring the outer circumferences (as reported in Supplementary Fig. 7). The measurements were set with the command Analyze > Set Measurements, and the "Perimeter" option was selected. Using the "Polygon selections" tool, the inner circumference was traced and measured. The radius was deduced from the measured circumference, and that value was used to calculate an approximate equatorial section area (or volume) for all the embryos analyzed under both control and p38-MAPKi conditions. The calculated areas in $\mu m^2$ (or volumes in picoliters (pl)) are plotted in scatterplots, with the means and standard deviations marked. The differences were statistically analyzed using the Mann–Whitney test, and the results, unless otherwise stated within individual graphs with specific $P$ values (if statistically nonsignificant), are marked according to the following significance intervals: $P < 0.0001$ (****), 0.0001–0.001 (***), 0.001–0.01 (**), and 0.01–0.05 (*).

**Quantitative real-time PCR (qRT-PCR)**. Embryos from each stated experimental condition were collected and immediately processed for RNA extraction and isolation using an ARCTURUS PicoPure RNA Isolation Kit (ThermoFisher Scientific; Cat. No. KIT0204) following the manufacturer's protocol. The entire eluted volume of total RNA was immediately DNase treated with TURBO DNase (ThermoFisher Scientific; Cat. No. AM2238) according to the manufacturer's protocol. The whole sample was then subjected to cDNA synthesis using SuperScript III Reverse Transcriptase (ThermoFisher Scientific; Cat. No. 18080044) as directed by the manufacturer with oligo d(T)16 (for Figs. 1f and 6c) (ThermoFisher Scientific; Cat. No. N8080128) or random hexamers (for Fig. 4c, d, f) (ThermoFisher Scientific; Cat. No. SO142), dNTP Mix (ThermoFisher Scientific; Cat. No. R0192) and RNase Inhibitor (ThermoFisher Scientific; Cat. No. N8080119). The synthesized cDNA was diluted as required with nuclease-free water, and 1 μl was used in 10 μl individual final SYBR Green-based qRT-PCR volumes (PCR Biosystems Ltd., London, United Kingdom; Cat. No. PB20.11). A Bio-Rad CFX96 Touch Real-Time PCR Detection System was used for data acquisition with the standard settings, and the initial analysis was performed with the accompanying Bio-Rad CFX Manager software. Triplicate measurements were obtained for each gene (the sequences of the individual oligonucleotide primers used at final concentrations of 300 nM are

provided in Supplementary Table 6) from up to three biological replicates that were each technically replicated. The averaged transcript levels of the analyzed genes were determined after internal normalization against *Tbp* or *H2afz* mRNA levels. The raw data were acquired and initially analyzed with CFX Manager and then processed in Microsoft Excel (for biological and technical replicate averaging) and GraphPad Prism 8 (for graphical output). The statistical tests used are detailed in Supplementary Data 2. Unless otherwise stated within individual graphs with specific $P$ values (if statistically nonsignificant), the following significance intervals were recognized: $P < 0.0001$ (****), 0.0001–0.001 (***), 0.001–0.01 (**), and 0.01–0.05 (*). The error bars denote the calculated standard deviations.

**Statistics and reproducibility**. All statistical tests conducted are detailed in Supplementary Data 2 and results thereof, if not mentioned within respective figures, are detailed in Supplementary Data 1 (see tabs corresponding to respective figures or supplementary figures). The data analyzed was typically from not less than 10 embryos per condition, and are detailed within respective figure legends. Details regarding biological and/or technical replicates are provided within respective methodological sections.

**Reporting summary**. Further information on research design is available in the Nature Research Reporting Summary linked to this article.

## Data availability
The RNA sequencing datasets are available in the GEO database with accession number GSE162233. The mass spectrometry proteomics data have been deposited to the ProteomeXchange Consortium via the PRIDE partner repository with the dataset identifier PXD025711.

## Code availability
No custom code was used. All software and code are available through the cited references.

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

## Acknowledgements

We acknowledge the Institute of Parasitology (Biology Centre of the Czech Academy of Sciences, in České Budějovice) for housing mice, Marta Gajewska (Institute of Oncology, Warsaw, Poland) and Anna Piliszek (Institute of Genetics and Animal Breeding, Polish Academy of Sciences, Jastrzębiec, Poland) for providing founder CBA/W mice, Alena Krejčí (Faculty of Science, University of South Bohemia, Czech Republic) for pooling resources and other members of our laboratory for providing valuable input and discussions. We further acknowledge Markéta Hančová (IAPG, Liběchov) for her help in carrying out experiments. We also acknowledge the core facility at Masaryk University, Brno, Czech Republic—Faculty of Informatics, supported by the MEYS CR (LM2018129 Czech-BioImaging), for assistance with image analysis. CIISB, Instruct-CZ Centre of Instruct-ERIC EU consortium, funded by the MEYS CR infrastructure project LM2018127, is gratefully acknowledged for financial support of measurements at the CEITEC Proteomics Core Facility. Computational resources were supplied by the project e-Infrastruktura CZ (e-INFRA LM2018140) provided within the program Projects of Large Research, Development, and Innovations Infrastructures. We also thank our peer reviewers for their insightful and helpful comments during the revision process. This work was supported by a project grant from the Czech Science Foundation/GACR (18-02891S), a Marie Curie Individual Fellowship (MSC IF 708255) awarded to L.G., and a Ph.D. student award given to P.B. by the Grant Agency of the University of South Bohemia (GAJU; 012/2019/P).

## Author contributions

P.B. and A.W.B. conceived the project, designed experiments, analyzed results, and wrote the manuscript. P.B. conducted experiments. P.B. and L.G. prepared samples for mass spectrometry and RNA sequencing. L.G. analyzed proteomic, phosphoproteomic, and transcriptomic data and wrote associated portions within the manuscript. T.M. and M.P. designed and produced scarce-sample polysome profiling fractions. A.H. analyzed images for blastocoel cavity volume determination. D.P. and Z.Z. performed phosphoproteomic mass spectrometry, performed preliminary data analysis, and wrote the associated methods. A.A. performed time-lapse imaging and analysis. D.J. and A.S. performed peripheral experiments.

## Competing interests

The authors declare no competing interests.
