## [Peer Review File · Communications Biology]

Reviewers' comments:

Reviewer #1 (Remarks to the Author):

Review of "p38-MAPK mediated rRNA processing and translation regulation enables PrE differentiation during mouse blastocyst maturation" by Bora et al

Studies focus on defining the mechanism of p38MAPK regulation of ICM specification and primitive endoderm differentiation (PrE) by applying live embryo imaging and proteomic and transcriptomic methods. Outcomes include demonstrating that pharmacological inhibition of p38-MAPK impairs blastocyst cavitation during a precise window of 3 hrs after treatment that also impairs PrE differentiation. Proteomic analysis shows downregulation of ribosomal proteins accompanied by upregulation of ribosomal mRNAs. Translation gene transcripts are especially affected by p38MAPK inhibition, including Mybpbp1a (rRNA transcription and processing regulator). siRNA downregulation of mybpbp1a is accompanied by reduced PrE. Outcomes include specificity of p38MAPK role in PrE as MEK/ERK inhibition does affect PrE and the p38MAPK inhibition effect is partially alleviated by mTOR activation.

The manuscript is well written and virtually free of any errors. It has been composed with care. The studies are similarly well planned and designed and the outcomes are presented appropriately. Overall, the study reports important outcomes that substantially increase our understanding of mechanism controlling cell fate acquisition during early development.

1. It is fascinating and important to have defined the window of p38MAPK regulation of blastocyst cavity size oscillation so precisely; I would predict this coincides with p38MAPK regulation of TE tight junction permeability and specifically TJ-actin cytoskeletal interactions.
2. The overall impact and timing must also be affected by treatment dose.....were other doses tested and were alterations to p38MAPK pathways constituents shown?
3. The targeted examination of developmental timing effects to MAPK mRNA levels of course does not show that the impaired cavity expansion is only due to effects to p38MAPK. The timing does correlate with a change in p38MAPK mRNA.
4. I do agree this is the first Mass spec based proteomic study of p38MAPK regulated proteins in the early embryo.
5. We would need some justification for the use of a 1.5-fold difference as an appropriate cut-off value criterion.
6. The transcriptomic analysis is also unique and to my knowledge has not been applied before to such precise timepoints. It is a bit odd that controls did not similarly show time related changes in transcript levels.
7. One factor that is not fully taken into consideration is that for both proteomic and transcriptomic analyses most of the cells in probed samples would be TE cells and not ICM which are the targets of interest for PrE differentiation.
8. Are all 6 selected targets expressed in ICM? I suspect they would also be expressed in TE.
9. The siRNA injection procedure into one 2-cell blastomere is state of the art and very appropriate.
10. The discovery of MYBBP1a as being implicated in p38MAPK cell fate mechanisms is also unique and important. Some of these experiments should likely be applied to mESCs to fully understand MYBBP1a's role in PrE. They are likely to be ongoing or important future steps.
11. The application of the o-propargyl-puromycin (OPP) assay is also state of the art. I do not know of it being applied to measure early embryo protein synthesis before. Ribosomal vs non ribosomal transcript fractionation is a well-established approach for determining translational activity in early embryos.
12. With the addition of HTATsfi and mTOR studies this becomes a very comprehensive study.
13. There are some thoughts that TE projections covering the ICM in the early blastocyst must be retracted allowing a free surface to form over the ICM and allowing PrE to form in the TE projections place as a cellular layer over the ICM. Perhaps the cavity contractions and expansions contribute to this process as well? It would be interesting to see if their retraction is impaired in p38MAPK inhibited blastocysts.
14. It appears that culture was under a 5% CO₂ in air atmosphere instead of a preferred 5/5/90

atmosphere. Was there a good reason for this?

15. The actual concentration of p38MAPK inhibitor is not provided early in the methods section. I have found it for the first time I believe on page 21 (20 uM SB220025) and of course Fig1. 20 uM SB220025 is a tricky dose as off target effects can accumulate at this dose.....were any other doses used? How was it decided to use 20uM? There is an inactive analogue called SB202474. It is a nice control I believe.

16. N values are not consistently presented in methods or all figure descriptions. Some certainly have them.

17. I very much like Figure 7 but I do not believe it is cited in the manuscript?

Reviewer #2 (Remarks to the Author):

The authors aim to understand the mechanisms by which p38 promotes primitive endoderm (PrE) formation in the mouse blastocyst. The authors show that inhibition of p38 shortly after E3.5 compromises PrE formation by E4.5 and leads to changes in the proteome, phosphoproteome, transcriptome, and in blastocyst cavity expansion en route to this developmental outcome. The authors highlight several candidate targets of p38, test them by knockdown for PrE phenotypes, and identify Mybbp1a as a regulator of pre-implantation development. They then show that translation, rRNA processing, and possibly transcription are negatively affected in p38 inhibition conditions. Finally, they show that inhibition of mTOR has a similar but not identical effect as p38 inhibition and that mTOR activation can partially rescue p38-inhibition.

A great strength of the study is the use of many different assays to explore the effect of p38 inhibition. The authors take a broad approach and should be commended for it. Weaknesses include the use of a single intervention to perturb p38 and the somewhat murky message that emerges from all of the different approaches. An overall theme of p38 -> mTOR -> translation -> PrE fate emerges, but the manuscript feels unfocused. The phosphoproteomics is novel and interesting, and the identification and knockdown of Mybbp1a and subsequent analysis is well done and interesting, but these experiments are tenuously connected to the subsequent figures and overall message of p38 -> mTOR -> translation. I am not sure if this manuscript should not be rewritten as two, removing the phosphoproteomics and Mybbp1a experiments for a future manuscript and focusing instead on the p38 -> mTOR -> translation axis. At the minimum, I strongly recommend streamlining the manuscript as much as possible.

We also found some specific issues that should be addressed prior to publication.

Issues

1. Lines 177-9. An alternative explanation is that Mapk13 and Mapk14 are redundant in this respect. If Mapk14 were the sole or dominant mediator of p38-dependent PrE maturation that in turn caused a failure to hatch in 40% of embryos, wouldn't one expect to see implantation defects or a reduction of Mapk14 mutants recovered at midgestation? Mapk14 mutants are recovered in normal ratios until E10.5 and are indistinguishable from littermates at E9.5, arguing for some sort of redundancy, whether through Mapk13 or some other mechanism at work in utero.

2. The phenotype appears to be somewhat later than with FGF/FGFR knock-outs, particularly comparing to the receptor knock-outs. This begs the question of whether p38 is acting downstream of FGFRs. There should also be a discussion of how p38 is engaged by FGFRs, as this is not entirely clear in the literature although FGF activation induces that of p38.

3. Lines 200-224. The authors mention the likelihood of direct and indirect targets of p38 in their phosphoproteomic dataset. Can they add commentary on whether any of the 51 reduced or mixed

targets were known p38 phosphorylation sites or if some were at least S/T sites followed by a Proline, suggesting a MAPK consensus motif?

4. Lines 380-390/ Figure 5 f-g. The authors posit an impaired transcriptional landscape based on staining for chromatin and RNAPoIII markers associated with active transcription, following a 10-hour inhibition of p38-MAPK. At this same timepoint, the authors' own RNA-Seq shows that transcription is only altered 480 genes overall. Of these, 272 are downregulated and 208 are upregulated. Is it the authors' contention that most genes are downregulated to an identical degree, such that the FPKM of most genes (other than the 480) is unchanged vs DMSO treatment, and that the total RNA output per cell is reduced, but that their DEGs are altered on top of this global reduction? If so, they should explicitly state this and provide some data indicating reduced total RNA per cell following MAPKi treatment. Otherwise, they must provide some explanation for these incongruous data before concluding there is a less permissive transcriptional landscape. We do note at that 10 hours the most downregulated genes are more highly downregulated than the most upregulated genes and (+7hr)), but this is not true at 7 hours and we are also unsure if this a large enough effect (at 10 hours) to be consistent with a globally less permissive transcriptional landscape.

5. Why not look at the p38/mTOR pathway in embryos that have undergone diapause?

Minor Issues

1. The discussion is overly long and at times speculative.

2. In figure 1c(iii) the label on the mutant trace claims that "few hatch" when the data in 1c indicate that 60% actually hatch. The use of "few" is misleading and should be softened e.g. to something like "no expansion and reduced hatching" or "no expansion and hatching defects"

3. Figure 2d. The use of "DNA" and "RNA" is confusing in the context of a proteomic study. Maybe alter to "DNA-Related Terms" and "RNA-Related Terms" to alert the reader these are all GO terms based on proteomics, not some other -omics dataset.

4. Fig 5e(i). The color of the SB220025 legend is different from the bar graph.

5. Line 504. "knock on" should be hyphenated "knock-on".

Author response to original reviewers' comments:

Reviewer #1 (Remarks to the Author):

Review of “p38-MAPK mediated rRNA processing and translation regulation enables PrE differentiation during mouse blastocyst maturation” by Bora et al

Studies focus on defining the mechanism of p38MAPK regulation of ICM specification and primitive endoderm differentiation (PrE) by applying live embryo imaging and proteomic and transcriptomic methods. Outcomes include demonstrating that pharmacological inhibition of p38-MAPK impairs blastocyst cavitation during a precise window of 3 hrs after treatment that also impairs PrE differentiation. Proteomic analysis shows downregulation of ribosomal proteins accompanied by upregulation of ribosomal mRNAs. Translation gene transcripts are especially affected by p38MAPK inhibition, including Mybpp1a (rRNA transcription and processing regulator). siRNA downregulation of mybpp1a is accompanied by reduced PrE. Outcomes include specificity of p38MAPK role in PrE as MEK/ERK inhibition does affect PrE and the p38MAPK inhibition effect is partially alleviated by mTOR activation.

The manuscript is well written and virtually free of any errors. It has been composed with care. The studies are similarly well planned and designed and the outcomes are presented appropriately. Overall, the study reports important outcomes that substantially increase our understanding of mechanism controlling cell fate acquisition during early development.

We thank the reviewer for their summary of our work and their appraisal of its quality, scope and relevance.

1. It is fascinating and important to have defined the window of p38MAPK regulation of blastocyst cavity size oscillation so precisely; I would predict this coincides with p38MAPK regulation of TE tight junction permeability and specifically TJ -actin cytoskeletal interactions.

We are inclined to agree with the reviewer's insight and have made changes to the revised *Discussion* section of the manuscript text to make this point clearer than it was in the original text (*i.e.* paragraph 2 of the *Discussion* section). Please note, major textual changes are highlighted in the submitted revised manuscript file (some sentences structures, *etc.* were edited or reordered but not highlighted in order not to over complicate the resubmitted file). We have also added extra references as appropriate.

2. The overall impact and timing must also be affected by treatment dose.....were other doses tested and were alterations to p38MAPK pathways constituents shown?

We have in the past published a paper describing preliminary findings on the impact of p38-MAPK inhibition (p38-MAPKi) on late mouse blastocyst derivation using the compound SB220025 (and its biologically inactive analog SB202474, as referred to by the reviewer below, as a negative control; plus, an alternative p38-MAPK inhibitor, SB203580) – see Thamodaran and Bruce 2016 (*Open Biology*). In this study, we did titrate the dose of SB220025 used and determined 20µM as the most effective concentration at providing the most complete phenotypic block in PrE differentiation across the blastocyst ICM (phenocopied using the related SB203580 compound but not observed using the inactive SB202474 analog). We therefore used this prior information to inform our choice of SB220025 concentration in the present study. Accordingly, we have made this point (and the point the reviewer makes relating to the explicit stating of the exact concentration used – see below) more clear in the revised *Results* and *Methods* sections of the manuscript. Additionally, the 20µM treatment dose used is in line with other studies of p38-MAPK function in the mouse embryo, in which similar titrations were performed (Natale *et al.*, 2004 – *Dev. Biol.*).

3. The targeted examination of developmental timing effects to MAPK mRNA levels of course does not show that the impaired cavity expansion is only due to affects to p38MAPK. The timing does correlate with a change in p38MAPK mRNA.

We agree with the reviewer's assessment and did not originally intend to give the impression the two were absolutely/unequivocally causal. We have therefore amended the relevant text in the revised manuscript to reflect the coincidental nature of the observations/measurements are merely consistent with this view.

4. I do agree this is the first Mass spec based proteomic study of p38MAPK regulated proteins in the early embryo.

As we are sure the reviewer will acknowledge, we were being deliberately circumspect in making this assertion but we are relieved they agree with us.

5. We would need some justification for the use of a 1.5-fold difference as an appropriate cut-off value criterion.

The reason for using the 1.5 fold cut-off, albeit under stringent statistical filtering (that was only applied in relation to the identification of differentially detected phospho-peptides in our phosphoproteomic screen – the mRNA-Seq and proteome DEGs were determined as described in the *Methods* section), was according to both literature precedent and a desire to

not miss potential p38-MAPK effectors; given the comparative lack of starting material we were limited by means we knew we were operating at the limits of mass spectrometry sensitivity. Moreover, it was for this reason we assayed the effect of clonal RNAi mediated expression knockdown of a selection of such candidates (*i.e.* six) to confirm mechanistic functionality (as significantly confirmed and described for the targeting of the *Mybbp1a* gene). We acknowledge the limitation but point to the coherence of the gene ontological analyses of DEGs identified in the transcriptome, general proteome and phospho-proteome (plus the functional RNAi mediated knockdown analyses) as evidence of the appropriateness of the employed cut-off.

6. The transcriptomic analysis is also unique and to my knowledge has not been applied before to such precise timepoints. It is a bit odd that controls did not similarly show time related changes in transcript levels.

We are again pleased to know the reviewer agrees with us about the uniqueness of our mRNA-Seq based transcriptomic analysis of mouse blastocysts in relation to p38-MAPK function.

We have to admit that we were also originally surprised to not observe significant changes in the general transcriptome of control treated embryo groups between the time-points assayed. On reflection, we think this most probably reflects the early stages of mouse blastocyst development (we assayed from E3.5 +4h, 7h & 10h – *i.e.* prior to the E4.0 mid-point of blastocyst maturation, defined by the emergence of the ‘salt and pepper’ pattern of mutually exclusive NANOG and GATA6 expressing EPI and PrE progenitors in the ICM) being primarily regulated (in terms of gene expression) on the proteomic level (be it by regulated translation, as uncovered here, and/or protein degradation – *i.e.* we know the formation of salt and pepper pattern of EPI or PrE progenitors requires the specific down-regulation of either NANOG or GATA6, respectively, from an initially uncommitted progenitor cell pool expressing both marker proteins). Hence, it is possible very little differential mRNA expression during this early blastocyst period may reasonably be expected. Indeed, we contend that it is the later blastocyst stages (post-E4.0), characterised by the differentiation of the specified PrE (during which the expression of hitherto non-expressed PrE marker genes is induced – *e.g.* *Sox17* and *Gata4*) and the maturation of the EPI (transiting, in an inter-cell competitive manner, towards optimal naïve pluripotency – Hashimoto and Sasaki, 2019 – *Dev. Cell*), that collectively represent the optimal developmental window to observe the most profound transcriptomic flux under unperturbed developmental conditions (as indeed shown in a clutch of previous studies – *e.g.* Ohnishi *et al.*, 2014 – *Nat. Cell Biol.*). We thus hypothesise our collective data suggest gene regulation on the proteome level is important for PrE specification during the early phase of blastocyst formation and acts as a necessary primer for transcriptionally driven differentiation during the later stages.

It is also important to note that we consider the profound changes we observed, across the three early blastocyst stages assayed, in the transcriptomes of p38-MAPK inhibited blastocysts (exhibiting a strong bias towards translation related genes), most probably reflect the induction of dysregulated feedback mechanisms resulting from the impaired overall translation at this time; rather than any ordinary process that would occur in unperturbed blastocyst development.

7. One factor that is not fully taken into consideration is that for both proteomic and transcriptomic analyses most of the cells in probed samples would be TE cells and not ICM which are the targets of interest for PrE differentiation.

We certainly acknowledge this point made by the reviewer and have endeavoured to address this in the revised manuscript text.

However, we would like to highlight that in our above mentioned preliminary study (Thamodaran and Bruce 2016 - *Open Biology*), plus a further follow-up study (assaying the strength of the p38-MAPKi mediated PrE differentiation phenotype under conditions of amino-acid starvation – Bora *et al.*, 2019 – *Front. Cell & Dev. Biol.*), we noted the administration of p38-MAPKi from E3.5 does not affect the expression of the TE marker protein CDX2; although less TE cells are observed. Indeed, even when the inhibitor is given from the early 8-cell stage, outer cells still uniquely express CDX2 despite the embryos arresting development at the 32-cell stage (when cultured to the equivalent E4.5 late blastocyst stage of similarly treated control embryos). These data suggest continuing TE specification is not critically sensitive to p38-MAPKi in a manner suggested by our observations of PrE differentiation. Nevertheless, this is not to exclude some functional aspect of TE function, that has a knock-on consequence for PrE differentiation, is not affected by p38-MAPKi and/or such a function works in concert with mechanism operational on the level of ICM cells (points further elaborated in the *Discussion* section).

Additionally, when designing these studies (primarily the empirical proteomic and transcriptomic screens after p38-MAPKi), we did consider comparing data obtained from intact blastocysts and ICMs isolated using an immuno-surgery based technique to remove the overlying TE. However, to obtain a sufficiency of starting material (\pm p38-MAPKi) for such screens (operating at the limits of sensitivity – particularly the phospho-mass spectrometry analyses) was not deemed technically feasibly at this time. Interestingly, there is evidence from the literature that shows mouse blastocyst ICMs isolated at E3.5 are able to form a differentiated superficial PrE layer after 24 hours in culture (Wigger *et al.*, 2017 – *Scientific Reports*), thus replicating the normal development of intact blastocysts. These data suggest that whilst TE cells (and potentially their associated p38-MAPK activities) may harbour a functional role in PrE differentiation, it is not necessarily essential and favour a more major role for ICM autonomous mechanisms.

Despite the above qualifications, we reiterate that we have amended the revised manuscript to better reflect our experiments were conducted on intact mouse blastocysts that comprised an intact TE and ICM.

8. Are all 6 selected targets expressed in ICM? I suspect they would also be expressed in TE.

We have consulted existing mouse blastocyst and post-implantation egg cylinder (E6.5) stage lineage specific transcriptomic data sets (*i.e.* Wang *et al.*, 2018 – *Nature Cell Biology*; describing blastocyst mRNA expression & Zhang *et al.*, 2016 – *Nature*; describing the transcript expression in the egg cylinder). We have now included this expression data in the supplementary information accompanying our revised manuscript. We find all 6 selected candidates are robustly expressed as mRNAs in both developmental stages. Moreover, this expression is largely equal between either the blastocyst TE and ICM or post-implantation stage epiblast (EPI) and extraembryonic ectoderm (ExE). These data indicate the expression of such genes is not subject to significant lineage specific transcriptional regulation (and functional regulation) and strengthens the argument their functional regulation is more likely to be post-transcriptional at the level of the proteome. We have revised the manuscript text to reflect these points, as highlighted by the reviewer's original comment.

9. The siRNA injection procedure into one 2-cell blastomere is state of the art and very appropriate.

We thank the reviewer for their endorsement of our approach to mechanistically probe the role of our identified candidate p38-MAPK effectors on mouse PrE blastocyst formation.

10. The discovery of MYBBP1a as being implicated in p38MAPK cell fate mechanisms is also unique and important. Some of these experiments should likely be applied to mESCs to fully understand MYBBP1a's role in PrE. They are likely to be ongoing or important future steps.

We thank the reviewer for sharing our view as to the importance of the role of MYBB1A in the observed p38-MAPKi mediated PrE phenotype of mouse blastocysts (particularly in relation to pre-rRNA processing and general translation).

As discussed in our manuscript, we were originally drawn to MYBBP1A (as identified in our phosphoproteomic screen for potential p38-MAPK effectors) by previous observations MYBBP1A is involved in regulating the level and processing of Pre-rRNAs (*e.g.* Hochstatter *et al.*, 2012 – *J. Biol. Chem.*, Tan *et al.*, 2012 – *J. Biomed. Sci.*) and is essential for early embryonic development (Mori *et al.*, 2012 – *PLoS One*). Furthermore, we considered these observations as highly consistent with the phenotypic and empirical screening results we obtained after p38-MAPKi in mouse blastocysts. Additionally, the combined confirmation that MYBBP1A is a known interactor of HTATSF1 (Hochstatter *et al.*, 2012 – *J. Biol. Chem.*) and HTATSF1 itself regulates mouse/human cell pluripotency (*via*, in part, regulation of pre-rRNA transcription and processing), in both *in vitro* ES cell embryoid models and *in vivo*, as shown in early post-implantation mouse embryos and especially pertinent to our study results (Corsini *et al.*, 2018 – *Cell Stem Cell*; a study from Juergen Knoblich's lab), strengthened our interpretation of the functional relevance of MYBBP1A in the observed blastocyst phenotype. It is therefore noteworthy that the majority of the work described by Corsini and colleagues was conducted using mouse and human ES cell lines (supported by latter observations in early post-implantation mouse embryos). We thus hypothesise, based on these reported results, that similar targeting of *Mybbp1a* in ES cells will most likely result in either a phenocopy or extremely similar results to those focussed on *Htatsf1*. Although we acknowledge that this is not a substitute for actually performing the experiments in ES cells but argue, in the described context, such experiments may lie outside the scope of our current study. Moreover, they may be better exploited by other groups (informed by our current study and those other referenced published reports) better placed to execute such ES cell based experiments.

11. The application of the o-proparyl-puromycin (OPP) assay is also state of the art. I do not know of it being applied to measure early embryo protein synthesis before. Ribosomal vs non ribosomal transcript fractionation is a well-established approach for determining translational activity in early embryos.

We again thank the reviewer for their endorsement of our combined approaches.

12. With the addition of HTATsfi and mTOR studies this becomes a very comprehensive study.

Having previously demonstrated the functional importance of functional p38-MAPK to blastocyst development and PrE differentiation (Thamodaran and Bruce 2016 - *Open Biology* & Bora *et al.*, 2019 – *Front. Cell & Dev. Biol.*) it was our stated aim in the current study to bring mechanism to our original observations. Hence, the use of empirical (phospho)proteomic and transcriptomic screens to uncover underlying and fundamental processes affected by p38-MAPKi (namely translation related) and identify relevant p38-MAPK effectors. We consider a comprehensive approach as required and necessary in this context – although in response to the second reviewers general comments, we have tried to make all of the same original points in our revised manuscript yet in a more streamlined manner (including rearranging the order of some of the presented results).

13. There are some thoughts that TE projections covering the ICM in the early blastocyst must be retracted allowing a free surface to form over the ICM and allowing PrE to form in the TE projections place as a cellular layer over the ICM. Perhaps the cavity contractions and expansions contribute to this process as well? It would be interesting to see if their retraction is impaired in p38MAPK inhibited blastocysts.

We agree with the reviewer's point regarding the retraction of TE projections (that overlay the ICM-blastocyst cavity interface) and the formation of the PrE (we assume they refer to classic TEM observation from Martin Johnson lab from the 1980s – although there have also been subsequent papers describing projections from the mural TE to the surface of the ICM in maturing blastocyst – suggested to perform some cell signalling role). We agree that it would indeed be interesting to observe if the retraction of such projections would be sensitive to inhibition of p38-MAPK activity. However, we hope the reviewer would not be offended that we feel the suggested experiment lies outside the scope of this current study, especially given their co-reviewer's thoughts on streamlining in the redrafted manuscript. However, it might interest the reviewer to know that the lab is currently optimising TEM techniques (to observe other blastocyst phenomena) and that, in light of this comment, we would now be open to observing such projections in the context of p38-MAPK function in the future.

14. It appears that culture was under a 5% CO₂ in air atmosphere instead of a preferred 5/5/90 atmosphere. Was there a good reason for this?

The reason the employed *in vitro* culture gas phase regime of 5% CO₂ in air (*i.e.* comprising 20% oxygen), rather than the so-called 5/5/90 regime (*i.e.* a mixture of 5% CO₂, 5% oxygen and 90% nitrogen – designed to more accurately reflect the physiological oxygen tension observed in the *in vivo* developmental setting of the oviduct/uterus), was to follow the well-established and standard precedents of the overwhelming majority of researchers in the field of mouse blastocyst lineage derivation. Thus, inherently permitting more readily comparable results. However, in all honesty we did not originally consider using a 5/5/90 regime and we acknowledge it would be of significant future interest to assay the effect of p38-MAPKi on blastocyst formation (and PrE differentiation) under these conditions. This is especially pertinent given our previous observation relating to increased detection of reactive oxygen species (ROS) in p38-MAPK inhibited blastocysts under conditions of amino acid depletion (Bora *et al.*, 2019 – *Front Cell & Dev. Biol.*). However, it would seem a little outside the scope of the current study (especially considering the requested streamlining from the second reviewer).

15. The actual concentration of p38MAPK inhibitor is not provided early in the methods section. I have found it for the first time I believe on page 21 (20 uM SB220025) and of course Fig1. 20 uM SB220025 is a tricky dose as off target effects can accumulate at this dose.....were any other doses used? How was it decided to use 20uM? There is an inactive analogue called SB202474. It is a nice control I believe.

We have edited the new manuscript draft to make the employed concentration of SB220025 more immediately obvious in the chronology of the written text (and to indicate it is based on previous reports in mouse preimplantation stage embryos). Regarding the employed doses and off target effects, plus utilisation of the inactive SB2020474 analog, we refer the reviewer to our response to point 2 above (reproduced for ease directly below) that we consider addresses these issues.

Reproduced authors' response to reviewer's original point 2:

We have in the past published a paper describing preliminary findings on the impact of p38-MAPKi on late mouse blastocyst derivation using the compound SB220025 (and its biologically inactive analog SB202474, as referred to by the reviewer below, as a negative control; plus, an alternative p38-MAPK inhibitor, SB203580) – see Thamodaran and Bruce 2016 (*Open Biology*). In this study, we did titrate the dose of SB220025 used and determined 20µM as the most effective concentration at providing the most complete phenotypic block in PrE differentiation across the blastocyst ICM (phenocopied using the related SB203580 compound but not observed using the inactive SB202474 analog). We therefore used this prior information to inform our choice of SB220025 concentration in the present study. Accordingly, we have made this point (and the point the reviewer makes relating to the explicit stating of the exact concentration used – see below) more clear in the revised *Results* and *Methods* sections of the manuscript. Additionally, the 20µM treatment dose used is in line with other studies of p38-MAPK function in the mouse embryo, in which similar titrations were performed (Natale *et al.*, 2004 – *Dev. Biol.*).

16. N values are not consistently presented in methods or all figure descriptions. Some certainly have them.

We have now modified the manuscript to include all N-numbers in the main figures and supplementary information.

17. I very much like Figure 7 but I do not believe it is cited in the manuscript?

Explicit reference to Figure 7 is made in the revised manuscript (under the section heading entitled *Conclusion*).

Reviewer #2 (Remarks to the Author):

The authors aim to understand the mechanisms by which p38 promotes primitive endoderm (PrE) formation in the mouse blastocyst. The authors show that inhibition of p38 shortly after E3.5 compromises PrE formation by E4.5 and leads to changes in the proteome, phosphoproteome, transcriptome, and in blastocyst cavity expansion en route to this developmental outcome. The authors highlight several candidate targets of p38, test them by knockdown for PrE phenotypes, and identify *Mybbp1a* as a regulator of pre-implantation development. They then show that translation, rRNA processing, and possibly transcription are negatively affected in p38 inhibition conditions. Finally, they show that inhibition of mTOR has a similar but not identical effect as p38 inhibition and that mTOR activation can partially rescue p38-inhibition.

We thank the reviewer for their accurate summary of our work. Please note, major textual changes (especially pertinent to the reviewers' collective comments) are highlighted in the submitted revised manuscript file (some sentences structures, etc. were edited or reordered but not highlighted in order not to over complicate the resubmitted file). We have also added extra references as appropriate.

A great strength of the study is the use of many different assays to explore the effect of p38 inhibition. The authors take a broad approach and should be commended for it. Weaknesses include the use of a single intervention to perturb p38 and the somewhat murky message that emerges from all of the different approaches. An overall theme of p38 -> mTOR -> translation -> PrE fate emerges, but the manuscript feels unfocused. The phosphoproteomics is novel and interesting, and the identification and knockdown of *Mybbp1a* and subsequent analysis is well done and interesting, but these experiments are tenuously connected to the subsequent figures and overall message of p38 -> mTOR -> translation. I am not sure if this manuscript should not be rewritten as two, removing the phosphoproteomics and *Mybbp1a* experiments for a future manuscript and focusing instead on the p38 -> mTOR -> translation axis. At the minimum, I strongly recommend streamlining the manuscript as much as possible.

We thank the reviewer for noting our comprehensiveness in attempting to further, in the face of significant experimental and technical limitations (as discussed below), our understanding of p38-MAPK function during mouse blastocyst maturation.

We acknowledge the raised weakness regarding the use of a single technical intervention to perturb p38-MAPK function. In mitigation we would like to highlight, from our own past experiences, other potential interventions open to us are not appropriate for the study of p38-MAPK in mouse preimplantation stage embryos. For example, and as referenced by the reviewer themselves (see main issues point 2), the use of genetic knockouts of the four individual *p38-Mapk* paralogous mouse genes do not result in preimplantation development phenotypes (or at least lethality – in fact it is only the loss of the *Mapk14* gene encoding the p38 α protein isoform that is eventually developmentally lethal). This indicates possible functional redundancy between expressed isoforms during preimplantation development (*i.e.* most likely MAPK13 and MAPK14, as discussed below) and could also be (partially) explained by maternal inheritance of masking functional p38-MAPK protein (unpublished observations of which we have obtained, using generic anti-p38-MAPK anti-sera, and that also render the use of microinjected siRNA mediated approaches, such as those presented in the current manuscript for the *Mybbp1a* gene, redundant). Hence, whilst not an ideal situation, we do consider our hands have been unavoidably tied in only being able to exploit the pharmacological inhibition experimental route. However, we would also like to draw attention to our previous study (Thamodaran and Bruce 2016 - *Open Biology*) in which we did microinject mRNA for a constitutively active form of the p38-MAPK activating kinase MKK6/MAP2K6 (p38-MAPKs represent the sole known substrates of MKK6/MAP2K6) into preimplantation stage mouse embryo blastomeres (*i.e.* both blastomeres at the 2-cell stage) and observed complete conversion of the ICM to a PrE fate (substantiating the importance of p38-MAPK activity to deriving the PrE lineage).

We have carefully considered the reviewer's suggestion to split the manuscript along the lines they suggest. However, we respectively disagree with the recommendation to omit the phosphoproteome/MYBBP1A data, as we feel the manuscript would then be, rightly in our view, criticised for being too descriptive and not furthering our mechanistic understanding of the role of p38-MAPKs in PrE derivation. It is for this reason that we have decided to retain this data in the current manuscript. However, we are not blind to the general criticisms of the reviewer regarding editing and in particular streamlining. We have made a significant structural change to the chronology of the *Results* and ended the manuscript with the phosphoproteome/MYBBP1A data, to improve the logical flow, and have endeavoured to slim the manuscript wherever possible (including the *Discussion* section); although we have also had to add extra data and clarifications at the collective request of the reviewers. We hope the changes will prove satisfactory to the reviewer.

We also found some specific issues that should be addressed prior to publication.

Issues

1. Lines 177-9. An alternative explanation is that *Mapk13* and *Mapk14* are redundant in this respect. If *Mapk14* were the sole or dominant mediator of p38-dependent PrE maturation that in turn caused a failure to hatch in 40% of embryos, wouldn't one expect to see implantation defects or a reduction of *Mapk14* mutants recovered at midgestation? *Mapk14* mutants are recovered in normal ratios until E10.5 and are indistinguishable from littermates at E9.5, arguing for some sort

of redundancy, whether through Mapk13 or some other mechanism at work in utero.

After considering the reviewer's remarks it is clear to us that this is a perfect example of the merits of the peer review process. We agree with the point made and its underpinning rationale. We have now made changes to the text that reflect our revised opinion (on the strength of the reviewer's insight); that the most likely explanation reflects functional redundancy between MAPK13 and MAPK14 proteins during blastocyst maturation.

2. The phenotype appears to be somewhat later than with FGF/FGFR knock-outs, particularly comparing to the receptor knock-outs. This begs the question of whether p38 is acting downstream of FGFRs. There should also be a discussion of how p38 is engaged by FGFRs, as this is not entirely clear in the literature although FGF activation induces that of p38.

In regard to the precise developmental timing at which p38-MAPK inhibition (p38-MAPKi) exerts its effect (regarding the PrE phenotype), we would argue this as a somewhat subjective question. Continuous inhibition of p38-MAPK from E3.5-E4.5 results in significantly impaired PrE formation (as determined by assaying GATA4 expression) with virtually all ICM cells expressing the EPI marker, NANOG (as we show here and have previously described – Thamodaran and Bruce 2016 - *Open Biology*, Bora *et al.*, 2019 – *Front. Cell & Dev. Biol.*); suggesting a late blastocyst phenotype due to the failed emergence of differentiating PrE monolayer by the late blastocyst stage. However, as described here and in the two previously mentioned studies, we also know such a regime of p38-MAPKi also results in a significant population of late blastocyst ICM cells that express both NANOG and GATA6 (with the remainder solely expressing the EPI marker, NANOG), reminiscent of the uncommitted state of all ICM cells prior to and around E3.5. Thus, such data suggest the p38-MAPKi phenotype is not late but rather earlier in blastocyst ICM maturation and specifically impairs the specification of PrE (but not EPI) from the initial uncommitted state. This interpretation is supported by our previous observation that p38-MAPKi after E4.0 does not elicit a defective PrE phenotype and results in normal ICM lineage restriction (Thamodaran and Bruce 2016 - *Open Biology*; in contrast to MEK1/2 inhibition). It is additionally supported by our current identification of a minimal early blastocyst window of p38-MAPKi, amounting to 3 hours (*i.e.* in the E3.5 +4h to +7h period), that can significantly impair differentiating PrE (GATA4+) cell numbers by the late (E4.5) blastocyst stage. Hence, we would argue the identified p38-MAPKi phenotype is early in terms of blastocyst maturation and ICM lineage specification/derivation and as we originally discussed is concomitant with published observations indicating the early blastocyst maturation phase (prior to PrE progenitor specification) as being the most sensitive to FGF4 mediated signalling (Bessonard *et al.*, 2017 – *Scientific Reports*); although PrE/EPI specification segregation overall occurs in an essentially asynchronous manner as the blastocyst matures (ensuring consistent ICM lineage composition - Saiz *et al.*, 2016 – *Nat. Commun.*).

Indeed, the phenotype of the *Fgf4* genetic knockout mouse blastocyst (as described in Kang *et al.*, 2013 – *Dev Cell*) is similarly early during blastocyst maturation; being typified by a failure to derive the mutually exclusive 'salt and pepper' expression pattern of EPI and PrE progenitors in the ICM, due to the initially unaffected expression of the PrE marker protein GATA6 (initially co-expressed with NANOG) not being sustained and thus leading to failed PrE differentiation (and a lack of subsequent *Gata4*/GATA4 and *Sox7*/SOX7 expression) by the late blastocyst stage (and an ICM comprised solely of NANOG+ EPI cells – as also seen in *Grb2* nulls – Chazaud *et al.*, 2006 – *Dev. Cell*). Additionally, comprehensive analysis of mouse blastocyst ICM lineage derivation in FGF receptor genetic knockouts (*i.e.* *Fgfr1* and *Fgfr2* – Kang *et al.*, 2017 – *Dev. Cell*) essentially reveal a similarly early blastocyst phenotype; typified by the failure of PrE progenitors (GATA6+ and NANOG negative) to emerge from an initially uncommitted ICM population (co-expressing both GATA6 and NANOG) and thus contribute the usual emergence of the salt and pepper pattern of EPI/PrE progenitors. For example, the double knockout of *Fgfr1* and *Fgfr2* is essentially a phenocopy of *Fgf4* and *Grb2* null blastocysts. Furthermore, the sole removal of the *Fgfr1* gene (expressed in all cells of the early blastocyst ICM) leads to a highly significant loss of PrE (defined by exclusive expression of GATA6 – somewhat similar to our observations after p38-MAPKi) and a late blastocyst stage ICM primarily composed of EPI cells (solely expressing NANOG, albeit at a higher level than control embryos; indicative of a trapped non-physiological state of naïve pluripotency); with the single *Fgfr2* knockout only exhibiting minor defects in the timing of PrE specification (suggestive of an ancillary role in refining ICM lineage allocation).

Thus, it could reasonably be argued that the phenotypes (in terms of the formation of specified and differentiating PrE) are not so temporally distinct between the published genetic *Fgf4*, *Grb2* and *Fgfr1/2* knockout models and the pharmacological p38-MAPKi approach (as present here and in our previous reports). However, we are cognisant that temporal coincidence of the phenotypes is far from definitive proof of FGF4 mediated activation of p38-MAPK; although it is true that we have in the past favoured such a mechanism of direct FGF4 mediated p38-MAPK activation (for example, we previously demonstrated that PrE deficits in the late blastocyst ICM caused by FGFR inhibition, alone and not in combination with MEK1/2 inhibition, could be overcome by activating endogenous p38-MAPK using a constitutively active recombinant MKK6/MAP2K6 construct, a p38-MAPK specific activating kinase; additionally, as the reviewer highlights, there is precedent from other cell systems for FGF ligands activating p38-MAPKs but with limited mechanistic details – now referenced in the revised manuscript). Thus, we have also been grappling with the central question currently posed by the reviewer; namely does p38-MAPK functional act downstream of liganded FGFRs in specifying and deriving differentiating PrE?

Therefore, during the period our manuscript was under review, we embarked on an experiment designed to address this point (albeit with its own limits and caveats). Accordingly, we sought to assay if the known pan-ICM conversion of cells to PrE cell-fate caused by supplementation of exogenous recombinant FGF4 (Frankenberg *et al.*, 2011 – *Dev Cell*, Kang *et al.*, 2013 – *Development*, Kang *et al.*, 2017 – *Dev. Cell*, Krawchuk *et al.*, 2013 – *Dev. Biol.* and Nichols *et al.*, 2009 – *Development*) could be blocked by p38-MAPKi. Hence, we treated 8-cell (E2.5) and early blastocyst (E3.5) stage embryos with FGF4 (+heparin) and cultured to the late blastocyst (E4.5) stage (\pm p38-MAPKi; given from E3.5, as earlier p38MAPK inhibition would cause arrested development at around 32-cells – see Thamodaran and Bruce 2016 - *Open Biology*) and assayed the ICM for NANOG (EPI) and GATA4 (PrE) expression. Contrary to our expectation, we observed p38-MAPKi was not able to block ICM conversion to a PrE cell fate (*i.e.* GATA4+ or at least NANOG negative cells) but p38-MAPKi in the absence of supplemented FGF4 was able to elicit the ordinarily observed impairment in PrE derivation. Interestingly, similar inhibition of MEK1/2 after supplementation of the same dose of exogenous FGF4 (both provided at the E3.5 early blastocyst stage) was able to block pan-ICM cell conversion to a PrE (GATA4+ and NANOG negative) fate, with all ICM cells solely expressing NANOG (adopting an EPI-like fate). These data suggest that the role of p38-MAPK in PrE specification/derivation may not be, unlike that of MEK1/2, downstream (or limited to the downstream consequences) of activated FGFRs (despite activation of p38-MAPK being able to rescue PrE formation during FGFR inhibition - Thamodaran and Bruce 2016 - *Open Biology*). However, we also have to caveat these data by the acknowledgment that the concentrations of exogenous FGF4 used (and informed from numerous previous studies; Frankenberg *et al.*, 2011 – *Dev Cell*, Kang *et al.*, 2013 – *Development*, Krawchuk *et al.*, 2013 – *Dev. Biol.* and Nichols *et al.*, 2009 – *Development*) are high (in our case 1 μ g/mL) and could be considered super-physiological (as reported in Frum and Ralston, 2020 – *Reproduction*).

Notwithstanding, this caveat we have decided to include this extra data in the revised manuscript as supplementary information accompanying our redrafted *Discussion*.

3. Lines 200-224. The authors mention the likelihood of direct and indirect targets of p38 in their phosphoproteomic dataset. Can they add commentary on whether any of the 51 reduced or mixed targets were known p38 phosphorylation sites or if some were at least S/T sites followed by a Proline, suggesting a MAPK consensus motif?

We have now completed an analysis of the primary amino acid sequence of the identified peptides displaying significantly depleted phosphorylation levels after p38-MAPKi (comprising 57 individual peptide sequences from 51 proteins, 6 of which reported two distinct depleted phosphopeptide sequences). We found 24 (42.1%) of these peptide sequences contained the [S/T]P consensus phosphorylation motif for proline-directed mitogen-activated-kinases in general (Bardwell, 2006 – *Biochem. Soc. Trans.* – new supplementary table S6b), suggesting they could well be direct targets of p38-MAPK activity (including the functionally characterised MYBBP1A protein). Additionally, we also consulted the ‘PhosphoSitePlus’ database (an on-line resource that catalogues all known and predicted protein phosphorylation sites for a range of model organisms, including mouse: www.phosphosite.org/homeAction - Hornbeck *et al.*, 2014 – *Nuc. Acids Res.*) to ascertain how many of the detected phosphorylation sites were characterised as being potentially mediated by p38-MAPKs and identified 14 protein matches (also see new supplementary table S6b). Accordingly, we have made reference to this new analysis in the revised text and compiled novel attendant supplementary information as part of the resubmission.

4. Lines 380-390/ Figure 5 f-g. The authors posit an impaired transcriptional landscape based on staining for chromatin and RNAPolIII markers associated with active transcription, following a 10-hour inhibition of p38-MAPK. At this same timepoint, the authors’ own RNA-Seq shows that transcription is only altered 480 genes overall. Of these, 272 are downregulated and 208 are upregulated. Is it the authors’ contention that most genes are downregulated to an identical degree, such that the FPKM of most genes (other than the 480) is unchanged vs DMSO treatment, and that the total RNA output per cell is reduced, but that their DEGs are altered on top of this global reduction? If so, they should explicitly state this and provide some data indicating reduced total RNA per cell following MAPKi treatment. Otherwise, they must provide some explanation for these incongruous data before concluding there is a less permissive transcriptional landscape. We do note at that 10 hours the most downregulated genes are more highly downregulated than the most upregulated genes and (+7hr)), but this is not true at 7 hours and we are also unsure if this a large enough effect (at 10 hours) to be consistent with a globally less permissive transcriptional landscape.

The reviewer is correct, in that we based our assertion p38-MAPKi is associated with an impaired transcriptional landscape on the evidence provided from our H4K16ac and RNAPol II S2p specific immuno-staining (new Fig. 4f & g). However, it was and continues to be impossible for us to ascertain global/general transcriptional changes from our existing mRNA-Seq data because our libraries were prepared without the necessary ‘spiked-in’ controls (as we originally did not anticipate a global transcriptional effect caused by p38-MAPKi). However, it is possible using the methods we describe to identify DEGs against any prevailing trend in overall general transcriptional output. In hindsight, it would have been beneficial to include such controls but we currently lack the necessary resources to repeat the mRNA-Seq experiments. However, we do know that in the related published study Bulut-Karslioglu and colleagues (Bulut-Karslioglu *et al.*, 2016 – *Nature*), studying mTORi induced blastocyst developmental diapause, similar H4K16ac and RNAPol II S2p derived data (also indicating reduced transcription, associated with mTORi) were associated with mRNA-Seq data (compiled from libraries with the requisite spiked-in controls) confirming reduced general transcriptional output. Thus, we feel confident that the H4K16ac and RNAPol II S2p specific immuno-staining data we present is indeed indicative of a generally impaired transcriptional landscape. We have attempted to make this point more clear in the revised manuscript.

5. Why not look at the p38/mTOR pathway in embryos that have undergone diapause?

We thank the reviewer for their suggestion. This is something that we have considered, or rather are considering, for future research but respectfully consider a little outside the scope of the current study centred on blastocyst maturation and PrE formation specifically (and a manuscript that we have been asked to streamline). However, we have quite extensively acknowledged the relevance of our findings in relation to diapause in our original and revised *Discussion*. The reviewer will know that an *in vitro* developmental diapause (mimicking *in vivo* diapause) can be induced in mouse blastocysts by chemical inhibition of mTOR (both mTORC1 and mTORC2 - Bulut-Karslioglu *et al.*, 2016 – *Nature*). Moreover, that we can (and report here) replicate such data. We would also note that in relation to ICM cell fate, the provision of mTOR inhibition from the early (E3.5) blastocyst stage is associated with blocked cell division (versus controls) and nearly all such cells remaining in an uncommitted state (expressing both GATA6 and NANOG; *i.e.* the eventual mutually exclusive marker proteins of the late blastocyst stage PrE and EPI, respectively). These combined data are consistent with *de facto* developmental diapause (*i.e.* as seen *in vivo*, a truly arrested development). However, p38-MAPKi during the same developing window significantly differs (as presented in our manuscript – new Fig. 5). Specifically, in that the ICM cell number continues to increase from E3.5 (although not at the same rate as control treated embryo groups) and EPI specification appears intact (as judged by ICM cells solely expressing NANOG and not GATA6). Moreover, successful PrE specification/differentiation (as judged by GATA4 expression) is significantly impaired, with the presumptive PrE progenitors blocked in the uncommitted state (*i.e.* co-expressing GATA6 and NANOG). Thus, it appears the effect of p38-MAPKi is restricted (at least in terms of the ICM) to PrE formation, rather than overall developmental diapause. Hence, we hypothesise p38-MAPK function may not play a major role in diapause *per se*, although we freely acknowledge this would need to be directly tested in the future (for example, by observing if mTOR inhibition is able to induce diapause in embryos with experimentally enhanced p38-MAPK activity - *e.g.* using pharmacological activators of p38-MAPK or over-expressing constitutively active MKK6/MAP2K6, the p38-MAPK activating kinase – as used in Thamodaran and Bruce 2016, *Open Biology*). Such ideas and experiments represent potential future research directions for our research group.

Minor Issues

1. The discussion is overly long and at times speculative.

In keeping with the reviewer's suggestion to streamline the manuscript we have significantly redrafted the *Discussion* section (*e.g.* removing the section relating to p38-MAPK and MNK1/2 substrates). Although this has proven difficult, given the need to add request clarifications, however we hope the reviewer will find the changes satisfactory.

2. In figure 1c(iii) the label on the mutant trace claims that “few hatch” when the data in 1c indicate that 60% actually hatch. The use of “few” is misleading and should be softened *e.g.* to something like “no expansion and reduced hatching” or “no expansion and hatching defects”

The reviewer is obviously correct and we acknowledge the oversight made in the original manuscript text. We have now amended the text to correct this misleading statement along the lines the reviewer suggested.

3. Figure 2d. The use of “DNA” and “RNA” is confusing in the context of a proteomic study. Maybe alter to “DNA-Related Terms” and “RNA-Related Terms” to alert the reader these are all GO terms based on proteomics, not some other -omics dataset.

We acknowledge the reviewer's criticism and have adopted their suggested terms in the revised manuscript.

4. Fig 5e(i). The color of the SB220025 legend is different from the bar graph.

We have made the necessary changes to make these components consistent.

5. Line 504. “knock on” should be hyphenated “knock-on”.

The suggested hyphenated term ‘knock-on’ has been adopted in the revised manuscript text.

Referee #1 is satisfied with the revision and made no comments to authors.

Reviewer #2 (Remarks to the Author):

The authors have addressed some of the points that we raised in our review. The response is very long-winded and takes quite a while to go through. I am not satisfied by their response to point #2, which is not at all a "subjective question" and goes to the heart of what pathway p38 is acting on in the blastocyst (RTKs that have been implicated include FGFRs, but also PDGFRs and EGFRs). The authors seem to be unfamiliar with most of the recent literature. There is good discussion of the FGF mutant phenotype in *Fgf4*^{-/-} mutants (PMID 23193166) and *Fgfr1*^{-/-}; *Fgfr2*^{-/-} mutants (PMID 28552559 and PMID 28552557 that they do not seem to know). The *Grb2*^{-/-} analysis performed by Chazaud et al. was not pushed to the same level (it is a much older story). Also, how p38 can be engaged downstream of the receptors deserves some attention and discussion, see PMID 27036966. Otherwise, the authors have decided to not consider some of our points so I leave the decision to the Editor's consideration.

Author responses to reviewers' comments on the redrafted manuscript

Referee #1 is satisfied with the revision and made no comments to authors.

We are pleased the reviewer found the manuscript acceptable for publication after our revisions.

Reviewer #2 (Remarks to the Author):

The authors have addressed some of the points that we raised in our review. The response is very long-winded and takes quite a while to go through. I am not satisfied by their response to point #2, which is not at all a “subjective question” and goes to the heart of what pathway p38 is acting on in the blastocyst (RTKs that have been implicated include FGFRs, but also PDGFRs and EGFRs). The authors seem to be unfamiliar with most of the recent literature. There is good discussion of the FGF mutant phenotype in *Fgf4*^{-/-} mutants (PMID 23193166) and *Fgfr1*^{-/-}; *Fgfr2*^{-/-} mutants (PMID 28552559 and PMID 28552557 that they do not seem to know). The *Grb2*^{-/-} analysis performed by Chazaud et al. was not pushed to the same level (it is a much older story). Also, how p38 can be engaged downstream of the receptors deserves some attention and discussion, see PMID 27036966. Otherwise, the authors have decided to not consider some of our points so I leave the decision to the Editor's consideration.

We apologise to the reviewer for not adequately addressing point #2 in our original response. We also thank the reviewer for highlighting the relevant recent literature, that has helped us to more readily understand and address the original point.

In the revised manuscript, we acknowledge that the *Fgf4* and *Fgfr1/2* gene knockout phenotypes do temporally precede the window of p38-MAPK activity (and translation regulation) we identified to be necessary for appropriate PrE formation. Moreover, that such *Fgf4* and *Fgfr1/2* genetic phenotypes are typified by initial expression of GATA6 in nascent ICM that is then not sustained during mouse blastocyst maturation (resulting in pan-ICM expression of EPI markers), which contrasts with the p38-MAPKi phenotypes we report (in which PrE deficits are typified by normal EPI specification but persistence of a population of NANOG and GATA6 co-expressing ICM cells; *i.e.* PrE progenitors that have failed to specify). We have made this point very clearly in the substantially reworked *Discussion* section.

Additionally, and together with the extra data we included attempting to block the pan-ICM PrE conversion caused by exogenous FGF4 (using pharmacological p38-MAPKi and MEK1/2i), we now clearly remark the PrE supporting role we have characterised for p38-MAPK is most likely independent of the classically described FGF4-FGFR1/2-ERK1/2 signalling axis. Furthermore, as guided by the reviewer, we have also expanded the *Discussion* to consider alternative mechanisms (with all the accompanying citations) of p38-MAPK activation important for PrE specification and differentiation; including non-canonical mechanisms involving FGFR2 (as suggested from multiple human *FGFR2* mutation induced pathophysiologies) and potentially integrating cooperative signalling via PDGFRa (itself linked to PrE survival via regulation of the mTOR pathway), plus a potential PrE specific mechanism predicated on activation of FGFR3 (although we also acknowledge the potential presence of other, as of yet, unrecognised/uncharacterised p38-MAPK activation mechanisms).

We hope that by substantially reworking the *Discussion* to take these specific points into account (and maintaining those already raised by both reviewers throughout the manuscript), the reviewer will be satisfied by our attempt to redress the originally unsatisfactory response to point #2. We hope they will find the revised manuscript suitable for publication.

We also want to highlight that in response to the reviewer's (and the editor's) original comments about streamlining and editing the manuscript to improve structure and readability, we have made additional large-scale changes to the text and employed the services of a professional editing service (an accompanying certificate verifying this service is included in this resubmission). We consider these changes have greatly improved the manuscript.

This document certifies that the manuscript

Commun. Biol. Bora et al., 2021

prepared by the authors

Pablo Bora, Lenka Gahurova, Tomáš Mašek, Andrea Hauserova, David...

was edited for proper English language, grammar, punctuation, spelling, and overall style by one or more of the highly qualified native English speaking editors at SNAS.

This certificate was issued on **April 2, 2021** and may be verified on the SNAS website using the verification code **07A8-0C7A-0731-E14A-F3CP**.

Neither the research content nor the authors' intentions were altered in any way during the editing process. Documents receiving this certification should be English-ready for publication; however, the author has the ability to accept or reject our suggestions and changes. To verify the final

SNAS edited version, please visit our verification page at secure.authorservices.springernature.com/certificate/verify.

If you have any questions or concerns about this edited document, please contact SNAS at support@as.springernature.com.

March 30, 2021

Dear Alexander Bruce,

Thank you for choosing Springer Nature Author Services. This manuscript, titled "p38-MAPK-mediated translation regulation during early blastocyst development is required for primitive endoderm differentiation in mice," is very interesting. The paper was edited for grammar, phrasing, and punctuation. In addition, many edits were made to further improve the flow and readability of the text. Below, we highlight the areas of this paper that we focused on in our edit.

Edits were made throughout your manuscript to ensure the correct use of punctuation, which helps to signal to readers how to read a sentence and thus greatly aids in comprehension.

Certain edits were made to remove redundant, repetitive or unnecessary phrasing and to present the information in a more straightforward manner.

Some edits were made to improve conciseness by trimming unnecessary words and streamlining the flow of your manuscript.

Comments were left if further clarification would be helpful or confirmation of the meaning of the text was necessary. Please review these comments and all our changes carefully for more detailed suggestions, as well as to ensure that the final version of the manuscript is fully accurate.

Thank you again for using our editing services; we wish you the best of luck with your submission.

Best regards,

Molly A.
Senior Editor
Springer Nature Author Services